# Carbon-coated MoS$_{1.5}$Te$_{0.5}$ nanocables for efficient sodium-ion storage in non-aqueous dual-ion batteries

Yangjie Liu[1,2], Xiang Hu[1,2], Junwei Li[2], Guobao Zhong[1,2], Jun Yuan[1,2], Hongbing Zhan[1✉], Yongbing Tang [3] & Zhenhai Wen [2✉]

Sodium-based dual-ion batteries have received increased attention owing to their appealing cell voltage (i.e., >3 V) and cost-effective features. However, the development of high-performance anode materials is one of the key elements for exploiting this electrochemical energy storage system at practical levels. Here, we report a source-template synthetic strategy for fabricating a variety of nanowire-in-nanotube MS$_x$Te$_y$@C (M = Mo, W, Re) structures with an in situ-grown carbon film coating, termed as nanocables. Among the various materials prepared, the MoS$_{1.5}$Te$_{0.5}$@C nanocables are investigated as negative electrode active material in combination with expanded graphite at the positive electrode and NaPF$_6$-based non-aqueous electrolyte solutions for dual-ion storage in coin cell configuration. As a result, the dual-ion lab-scale cells demonstrate a prolonged cycling lifespan with 97% capacity retention over 1500 cycles and a reversible capacity of about 101 mAh g$^{-1}$ at specific capacities (based on the mass of the anode) of 1.0 A g$^{-1}$ and 5.0 A g$^{-1}$, respectively.

[1] College of Materials Science and Engineering, Fuzhou University, Fuzhou 350108, P. R. China. [2] CAS Key Laboratory of Design and Assembly of Functional Nanostructures, and Fujian Provincial Key Laboratory of Nanomaterials, Fujian Institute of Research on the Structure of Matter, Chinese Academy of Sciences, Fuzhou, Fujian 350002, P. R. China. [3] Functional Thin Films Research Center, Shenzhen Institutes of Advanced Technology, Chinese Academy of Sciences, Shenzhen 518055, P. R. China. ✉email: hbzhan@fzu.edu.cn; wen@fjirsm.ac.cn

L ithium-ion batteries (LIBs) play important roles in modern society yet face challenges of the growing cost of lithium-based resources and uneven geographical distribution[1],[2]. Thus, it is widely accepted that sodium-ion batteries (SIBs) are promising alternatives to LIBs, especially in the grid-scale energy storage due to the abundant source and low cost of sodium[3]. Among the emerging sodium-ion energy storage devices, sodium-based dual-ion batteries (SDIBs) with graphite as both anode and cathode have attracted increasing research interests thanks to their high operating voltage[4]. However, the current SDIBs are still unsatisfactory to meet the requirement of high energy density, because graphite has a relatively low capacity of around 30 mAh g$^{-1}$ for $NaC_{64}$ as anode and about 112 mAh g$^{-1}$ for $C_{20}PF_6$ as cathode[5]. Especially for the anode, soft carbon and hard carbon have been widely studied as the anode of SDIBs while showing a relatively low capacity so far[6],[7]. Alloys and metal oxides with enhanced capacity have thus been investigated but tend to undergo extra volume expansion during sodiation/de-sodiation with unsatisfactory cycling stability[8],[9]. In this context, we still need to strive to explore appropriate anode materials with high capacity and stable structures, which is of great importance to push forward the prosperity of SDIBs.

Molybdenum disulfide ($MoS_2$), as a classical two-dimensional lamellar structure, is apt for $Na^+$ storage as it has a tunnel space of ~0.62 nm and a theoretical capacity of 670 mAh g$^{-1}$ [10–13]. Nevertheless, $MoS_2$ also faces several critical issues including low electronic conductivity, possible aggregation of $MoS_2$ nanosheets, and considerable volume variations upon repeated charge–discharge process, which greatly restrict their electrochemical performance[14–16]. Recently, anion doping has been reported as a promising method to modify the physicochemical properties of nanomaterials and thereby potentially improve the associated electrochemical performance[17–20]. Tellurium (Te) element, as one member of the chalcogens group, has a lower electronegativity and forms more polarizable ions than S thanks to its larger atomic radius and enhanced shielding effect, which makes the valence electron of the Te atom less restrained by the nuclear charge[21]. Thus, the Te tend to enhance the conductivity of $MoS_2$ when used as dopants into $MoS_2$ by narrowing interlayer band gaps of $MoS_2$ with manifesting metallic properties; meanwhile, the weak Mo-Te band is easier to break during the conversion reaction than Mo-S band, which can effectively boost electrochemical process kinetics[22–24].

In terms of structural design, one dimensional (1D) wire-in-tube structure is a favorable architecture that could help address or circumvent the issues in anode materials of SDIBs. In particular, (1) the 1D wire-in-tube structure with a hollow cavity can provide an additional interfacial area for electrode/electrolyte contact; (2) such structure could accommodate volume expansion and maintain structural stability; and (3) the nanowire inside each nanotube not only increases the loading of active materials but also builds a fast electronic pathway for electrons transfer. Unfortunately, compared to the conventional and complex multi-step synthesis of 1D hierarchical nanotube materials, a simple and general strategy for the preparation of "1D nanowire-in-nanotube structure" with anion-doping is still a great challenge and has not been reported[10],[25].

Bearing the abovementioned points in mind, herein, we reported a source-template strategy to produce nanowire-in-nanotube structures with in situ-grown carbon film coating ($MoS_{1.5}Te_{0.5}$@C) termed as nanocables. In these structures, Te nanotubes are used as template and Te source for doping, thus, combining the desired features of adequate electronic conductivity and expanded interlayers distance (i.e., 0.75 nm). The systematic studies carried out clearly show the ability of the $MoS_{1.5}Te_{0.5}$@C nanocables to store Na ions in a non-aqueous battery system allowing for efficient cycling at high specific currents (i.e., >1 A g$^{-1}$), good cycling stability, and adequate rate capability performance. When the $MoS_{1.5}Te_{0.5}$@C nanocables are tested as negative electrode active material in combination with expanded graphite (EG)-based positive electrode with a $NaPF_6$-containing non-aqueous electrolyte solution, the dual-ion $MoS_{1.5}Te_{0.5}$@C‖EG cell deliver an operative voltage of about 3.1 V at 1 A g$^{-1}$.

## Results

**Materials synthesis and characterization.** Te nanomaterials (e.g., nanorod and nanowire) have been widely used as templates for fabricating a variety of tubular nanomaterials, yet there is the little report for nanocables synthesis. In the previous works, the targeted materials tend to grow on the outside surface of the Te nanowire/nanotube, likely due to the diffusion limit of ions under the conventional hydrothermal process[26]. The stirring assistance during the hydrothermal reaction can facilitate mass transportation and thus increase reaction rate as well as the flowability of solution[27]. Therefore, it is possible for the active ions to permeate into the inner space of the Te nanotube using the stirring-hydrothermal method, which ensures the formation of wire-in-tube $MoS_{1.5}Te_{0.5}$@C nanocables. The formation mechanism of nanowire-in-nanotube $MoS_{1.5}Te_{0.5}$@C nanocables is schematically illustrated in Fig. 1a and Supplementary Fig. 1. Te nanotubes (Supplementary Fig. 2) as the template were evenly dispersed in the mixed solution containing sodium molybdate, thiourea, and glucose, the subsequent stirring-hydrothermal process induce the in situ growth of carbonaceous film modified $MoS_2$ nanosheets on both inner and outer surface of Te nanotubes (Te@C/$MoS_2$, Supplementary Fig. 3). The nanowire-in-nanotube structured $MoS_{1.5}Te_{0.5}$@C nanocables can be finally evolved by annealing Te@C/$MoS_2$ in $H_2$/Ar atmosphere, during which Te nanotube templates are vaporized as Te source ($H_2$Te) for partially substitution of S in $MoS_2$ to form $MoS_{1.5}Te_{0.5}$ with carbon film decoration.

The morphology and structure of the as-prepared products were first examined by field-emission scanning electron microscopy (FESEM) and transmission electron microscopy (TEM). According to the SEM images in Fig. 1b, one can see that all samples have a typical 1D structure with a length of around ~8 µm and a diameter of around ~400 nm, indicating the well maintaining of 1D structure upon hydrothermal growth and the further calcination process. A close observation of $MoS_{1.5}Te_{0.5}$@C structure reveals the characteristic nanocables morphology of the material, in which the outer surface is abundant wrinkled nanosheets and one nanowire inside each nanotube is also constructed by nanosheets building blocks (Fig. 1c). Such unique structure is further validated by TEM images, as shown in Fig. 1d, e, in which one can observe that a single nanowire with a diameter of ~50 nm is almost coaxial with each nanotube. Importantly, the high-resolution TEM (HRTEM) image (Fig. 1f) reveals that the $MoS_{1.5}Te_{0.5}$ nanosheets are few-layers structure (2–5 layers) with an interlayer spacing of 0.75 nm, which is 20% larger than that (0.62 nm) of pristine $MoS_2$, indicating that the partial substitution of S in $MoS_2$ with larger-size Te can enlarge the lattice spacing. The aberration-corrected high-angle annular dark-field scanning TEM (STEM, Fig. 1g) image of $MoS_{1.5}$-$Te_{0.5}$@C nanocables reveals that there are a few brighter spots occupying some anionic sites with distinguishable contrast, corresponding to the heavier and larger Te atoms. Besides, it can be observed that several anions have vanished in the selected regions (as marked by the white circles), confirming the successful introduction of anionic defects[28]. Meanwhile, STEM energy dispersive spectroscopy (STEM-EDS) mapping (Fig. 1h, i

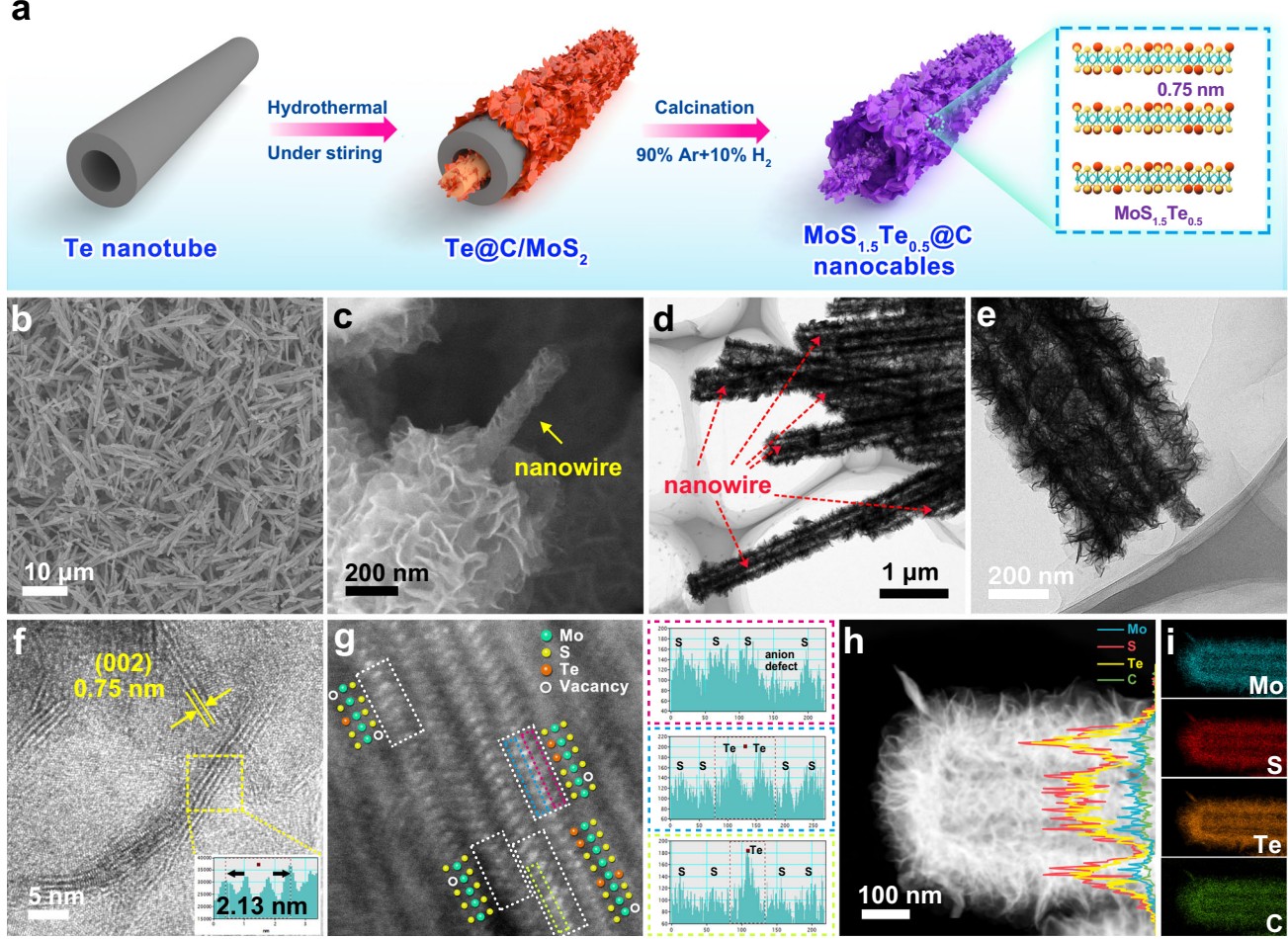

**Fig. 1 Morphology and structural characterization of MoS$_{1.5}$Te$_{0.5}$@C nanocables. a** Schematic illustration for the synthesis process of the MoS$_{1.5}$Te$_{0.5}$@C nanocables; **b**, **c** SEM images and **d**, **e** TEM images of the MoS$_{1.5}$Te$_{0.5}$@C nanocables. **f** High-resolution TEM image and **g** the magnified high-resolution STEM image for MoS$_{1.5}$Te$_{0.5}$@C nanocables (left) and the intensity profile of the colored areas indicated in the image (right). **h**, **i** HAADF-STEM and elemental mapping images of the MoS$_{1.5}$Te$_{0.5}$@C nanocables. The inset of panels (f) is the intensity profile along the lines taken from the lattice fringes based on HRTEM images, and the inset in (i) is line scan curves.

and Supplementary Fig. 4) confirms a successful and uniform Te doping in MoS$_{1.5}$Te$_{0.5}$@C nanocables with a Te/S ratio of ~1:3. Besides, the inductively coupled plasma optical emission spectroscopy (ICP-OES) results manifest that the Te:S ratio value is around 1:3 for Te-doped samples, respectively, which are basically consistent with the EDS results (Supplementary Table 1). Additionally, Mo, Te, S, and C signals in line-scan analysis and EDS mapping are uniformly distributed in both the core and the shell of the cables, further demonstrating that both the inner nanowire and the outer nanotube are MoS$_{1.5}$Te$_{0.5}$@C. For comparison, Te nanorods were also prepared as the source-template to fabricate MoS$_{1.5}$Te$_{0.5}$@C nanotubes rather than nanocables, which show a nanotube structure with enlarged-layered MoS$_{1.5}$Te$_{0.5}$ nanosheets as building blocks (Supplementary Fig. 5). In addition, carbon nanotube (CNT) was applied as the substrate to grow MoS$_2$ (CNT@MoS$_2$), which shows a core-shell structure with uniform MoS$_2$ nanosheets coating on CNT (Supplementary Fig. 6). It is noted that the interlayer spacing (0.62 nm) of the CNT@MoS$_2$ samples is not expanded; while for MoS$_{1.5}$Te$_{0.5}$@C nanotubes, one can also observe an increased interlayer distance of 0.73 nm. These results further reveal the incorporation of Te atoms in MoS$_2$ can effectively enlarge interlayer distance. More interestingly, this method can be extended to a general approach for the preparation of the other

wire-in-tube structural metal sulfide with Te-doping, including wire-in-tube WS$_x$Te$_y$@C nanocable and wire-in-tube ReS$_x$Te$_y$@C nanocable (Supplementary Fig. 7), which further highlight the novelty in synthetic strategy.

X-ray powder diffraction (XRD) measurements were carried out to characterize the crystalline phase of the set of samples, as shown in Fig. 2a. All three samples show a set of characteristic reflections that are well in agreement with the hexagonal 2H-phase of MoS$_2$ (JCPDS No. 37-1492)[29]. The CNT@MoS$_2$ nanotubes exhibit a higher intensity in major diffractive reflections than the MoS$_{1.5}$Te$_{0.5}$@C nanotubes and the MoS$_{1.5}$-Te$_{0.5}$@C nanocables, likely due to fewer defects in the former sample. Moreover, one can clearly observe that the (002) diffractive reflection for MoS$_{1.5}$Te$_{0.5}$@C samples obviously shift negatively toward a lower degree relative to that of CNT@MoS$_2$ nanotubes (Supplementary Fig. 8), implying that the interlayer spacing is expanded due to the introduction of large-sized Te into MoS$_2$ lattice, which is well consistent with the TEM observation. The phase evolution from MoS$_2$ to MoS$_{1.5}$Te$_{0.5}$ was also investigated by Raman spectra (Fig. 2b). There are two prominent peaks at ~376.2 and ~399.3 cm$^{-1}$ in the CNT@MoS$_2$ samples, corresponding to the in-plane Mo-S $E_{2g}$ vibration and out-plane Mo-S $A_{1g}$ vibration mode[30]. These two peaks shift negatively to low frequency because of the different coordination relations

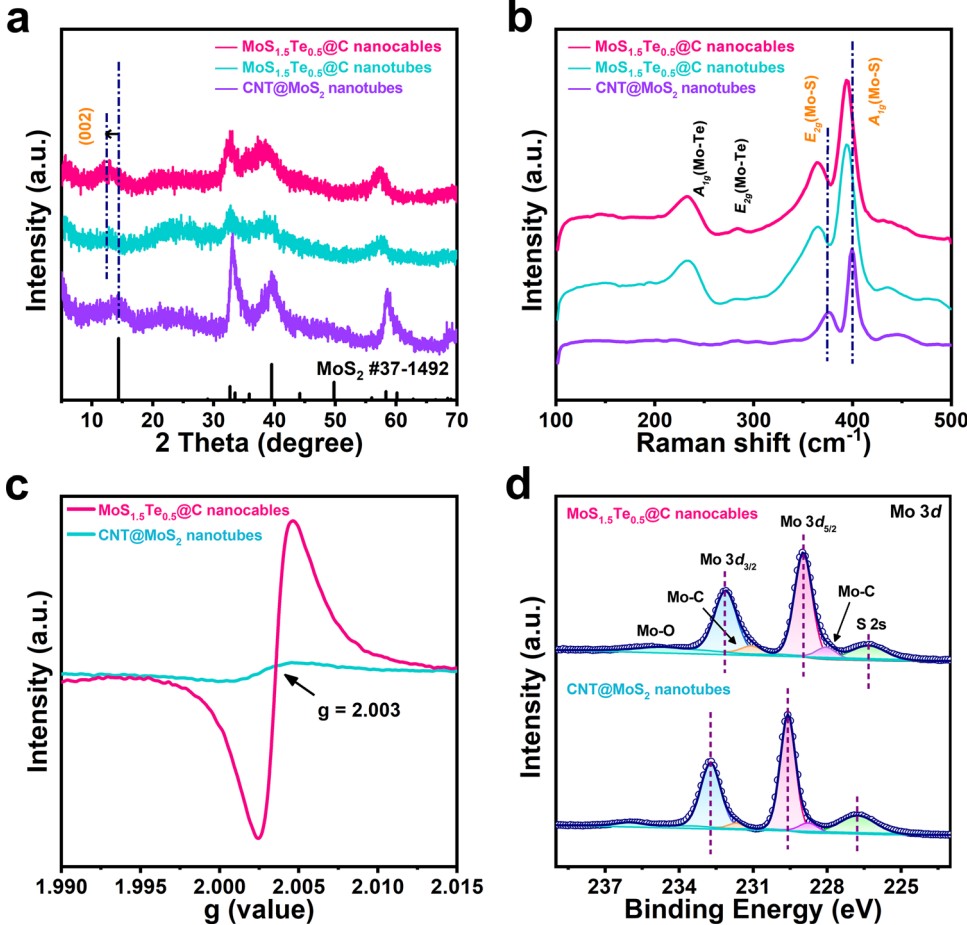

**Fig. 2 Characterization of the Mo-containing active materials. a** XRD pattern, **b** Raman spectra, and **c** EPR results for the Mo-containing active materials. **d** High-resolution XPS spectrum of Mo 3d for the $MoS_{1.5}Te_{0.5}$@C nanocable and CNT@$MoS_2$ nanotube composites.

between S/Te and Mo[22]. Meanwhile, two new Raman peaks associated with in-plane Mo-Te $E_{2g}$ (283.1 cm$^{-1}$) and out-plane Mo-Te $A_{1g}$ (232.5 cm$^{-1}$) vibrational mode emerge, additionally proving the successful replacing of partial S with Te[31].

Electron paramagnetic resonance (EPR) tests are performed to study the defect structure of the set of samples (Fig. 2c). The CNT@$MoS_2$ sample shows a quite weak signal at the g-factor of 2.003, nevertheless, the $MoS_{1.5}Te_{0.5}$ samples almost showed one-order magnitude higher signal of intensity than that of CNT@$MoS_2$ thanks to the increased anionic defects by Te doping in $MoS_{1.5}Te_{0.5}$, it is thus rational that the introduction of Te atoms plays a key role in increasing the defect concentration. X-ray photoelectron spectroscopy (XPS) measurements are utilized to further study the electronic states surface and chemical composition for the products (Fig. 2d and Supplementary Figs. 9–11). The survey spectra identify the co-existence of S, Te, C, Mo elements in $MoS_{1.5}Te_{0.5}$@C nanocables (Supplementary Fig. 9). Figure 2d shows the high-resolution Mo 3d XPS spectrum of the $MoS_{1.5}Te_{0.5}$@C nanocables and the CNT@$MoS_2$ nanotubes, where one can observe two peaks at ~232.1 and ~229.0 eV relating to $3d_{5/2}$ and $3d_{3/2}$ for Mo$^{4+}$, accompanied with a pair of shoulder peak at ~231.0 and ~228.3 eV referring to Mo-C bond[32,33]. The covalent interaction formation of Mo-C bond indicates the carbonaceous polymer-derived carbon film tightly wrap on the surface of $MoS_2$ that preferably enhances the electrical conductivity. In addition, the signal peaks of both Mo 3d and S 2p (Supplementary Fig. 10a) for the $MoS_{1.5}Te_{0.5}$@C nanocables show an obvious blue shift to lower binder energy, signifying the formation of Mo–Te bonds can effectively modify

the redox states of the Mo and S atoms by redistributing electronic structure[23,34]. For the high-resolution Te 3d spectrum in the $MoS_{1.5}Te_{0.5}$@C nanocables (Supplementary Fig. 10b), the peaks at 572.8 and 583.1 eV are related to Te $3d_{5/2}$ and Te $3d_{3/2}$ arising from Mo–Te bonds in $MoS_{1.5}Te_{0.5}$@C. Besides, two satellite peaks at 576.0 and 586.5 eV can be observed, which are likely arisen from the oxidized-Te species because of slight oxidation (TeO$_2$) exposed in the air[35,36]. It should be pointed that the corresponding peaks (Mo and S) in the $MoS_{1.5}Te_{0.5}$@C nanotubes also show a blue shift (Supplementary Fig. 11), further verifying the doping effectiveness. Taking together, these results further verify the phase transition from $MoS_2$ to $MoS_{1.5}Te_{0.5}$ with the presence of strong covalent bond interaction between Mo and Te/S, accompanied by the increase in defect sites, which can be greatly beneficial for Na$^+$ storage. The weight content of carbon in the $MoS_{1.5}Te_{0.5}$@C nanocables, the $MoS_{1.5}Te_{0.5}$@C nanotubes, and the CNT@$MoS_2$ nanotubes samples are about ~9.1%, ~12.1%, and ~15.9%, respectively, according to thermogravi-metric analysis (TGA, Supplementary Fig. 12). Additionally, the mass proportion of Te-doping in $MoS_{1.5}Te_{0.5}$@C nanocables is as high as ~30.7 wt%. The Brunauer-Emmett-Teller (BET) specific surface area and corresponding pore-size distribution were estimated based on the N$_2$ adsorption/desorption tests (Supple-mentary Fig. 13 and Supplementary Tables 2 and 3). The specific surface area of $MoS_{1.5}Te_{0.5}$@C nanocables is 21.9 m$^2$ g$^{-1}$, which is less than those of the $MoS_{1.5}Te_{0.5}$@C nanotubes (24.8 m$^2$ g$^{-1}$) and CNT@$MoS_2$ nanocables (36.7 m$^2$ g$^{-1}$). The lower surface area for $MoS_{1.5}Te_{0.5}$@C nanocables is attributed to the heavier Te-atoms doping.

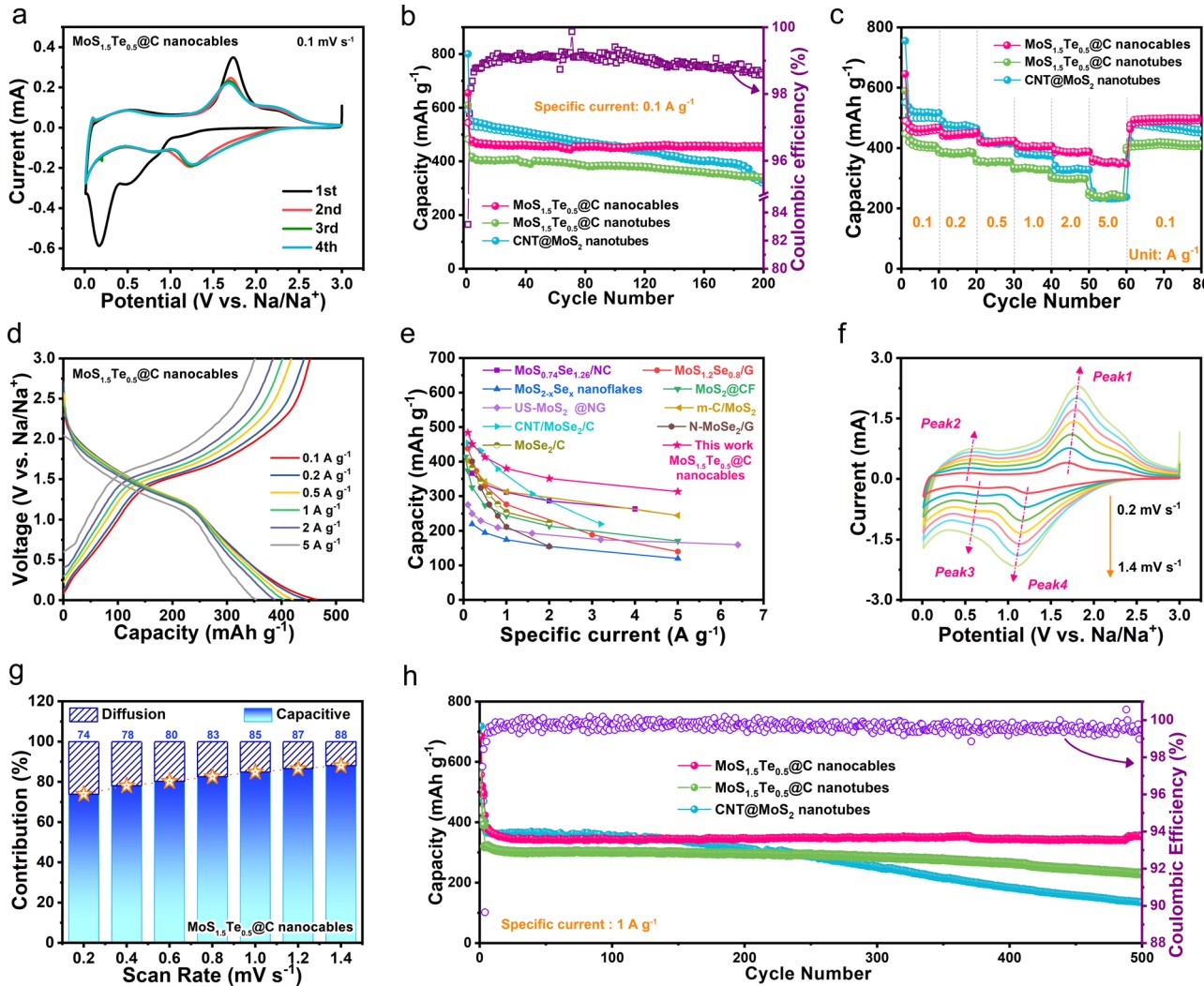

**Fig. 3 Electrochemical characterizations of the Mo-containing active materials. a** CV curves of $MoS_{1.5}Te_{0.5}$@C nanocables electrode at a sweep rate of 0.1 mV s$^{-1}$. **b** Cycling performance, **c** rate capabilities test, and **d** corresponding charge/discharge curves at various specific currents for the three Mo-containing active materials using the 1 M NaPF$_6$ in EC/EMC/DMC (1:1:1, by volume) electrolyte solution at 25 °C. **e** Comparison of sodium-ion storage performance of $MoS_{1.5}Te_{0.5}$@C nanocables electrode with previously reported Mo-based electrode materials. **f** CV curves at different scan rates and **g** the percentages of capacitive and diffusion-controlled capacities at different scan rates in the $MoS_{1.5}Te_{0.5}$@C nanocables electrode. **h** Comparison of long-term cycling stability at 1.0 A g$^{-1}$ for the three different Mo-containing active materials investigated.

**Electrochemical energy storage characterizations**. A 2032 half-cell was first assembled by pairing a working electrode with the Mo-containing materials mentioned above, a Na metal foil as reference and counter electrode, and a NaPF$_6$-based non-aqueous electrolyte solution. Figure 3a presents the cyclic voltammetry (CV) curves of the $MoS_{1.5}Te_{0.5}$@C nanocables-based cell at a scan rate of 0.1 mV s$^{-1}$. During the first sweep, one weak cathodic peak at around 0.9 V is observed, corresponding to the insertion reaction of Na$^+$ into the $MoS_{1.5}Te_{0.5}$ layer with forming Na$_{x}$-$MoS_{1.5}Te_{0.5}$. The evident cathodic peaks at about 0.5 and 0.2 V can be ascribed to the formation of a solid electrolyte interphase (SEI) and corresponding conversion reaction from Na$_x$MoS$_{1.5}$Te$_{0.5}$ to metallic Mo nanoparticles, Na$_2$S and Na$_2$Te[37]. The oxidation peak in the reversed scan at 1.8 V is assigned to the corresponding reverse reaction[38,39]. In the following three scans, the almost overlapped CV curves indicate good reversibility and stability of the $MoS_{1.5}Te_{0.5}$@C nanocables electrodes. Supplementary Figure 14 exhibits the initial four cycle's galvanostatic charge-discharge (GCD) curves at 0.1 A g$^{-1}$, where the corresponding voltage platforms are well in agreement with the redox behavior in

CV results. Furthermore, the $MoS_{1.5}Te_{0.5}$@C nanocables electrode shows reversible and durable cycling stability at 0.1 A g$^{-1}$ (Fig. 3b). The $MoS_{1.5}Te_{0.5}$@C nanocables electrode delivers an initial discharge and charge capacities of 655.2 and 544.8 mAh g$^{-1}$ with an initial Coulombic efficiency (ICE) 83.2%, which are lower than those (800.3/591.8 mAh g$^{-1}$) of CNT@MoS$_2$, but the former show considerably higher ICE than the latter (73.9%). In order to exclude the influence of carbon films, we carried out experiments to prepare the pure $MoS_{1.5}Te_{0.5}$ and MoS$_2$ samples without carbon coating and evaluate their ICE for Na$^+$ storage. The $MoS_{1.5}Te_{0.5}$ electrodes show an average ICE of ~88.4%, this value is higher than that of MoS$_2$ sample (~74.8%, Supplementary Fig. 15). Thus, the enhanced ICE of $MoS_{1.5}Te_{0.5}$@C nanocables can be rationally attributed to improved reversibility of the sodiation/desodiation process, since the weaker interaction of Mo–Te bonds was beneficial for the conversion reaction than that of Mo–S bands, which favor for lowering reaction barrier with improved reversibility. The irreversible capacity loss in the first sodiation/desodiation cycle of the $MoS_{1.5}Te_{0.5}$@C-based electrode could be attributed to the doping of the heavy Te atoms.

Figure 3c compares the rate performance of the three different Mo-containing active materials (i.e., $MoS_{1.5}Te_{0.5}@C$ nanocables, $MoS_{1.5}Te_{0.5}@C$ nanotubes and $CNT@MoS_2$ nanotubes). The $MoS_{1.5}Te_{0.5}@C$ nanocables can release reversible capacities of 462.3, 447.7, 421.0, 407.5, 387.1, and 352.3 mAh g$^{-1}$ at 0.1, 0.2, 0.5, 1, 2, and 5 A g$^{-1}$, respectively. When the specific current re-switches back to 0.1 A g$^{-1}$, the capacity could recover its original capacity level, the performance parameters in terms of stability and rate capability are better than those of $MoS_{1.5}Te_{0.5}@C$ nanotubes and $CNT@MoS_2$ nanotubes (Supplementary Fig. 16). Particularly, we can observe that the plateaus in GCD curves (Fig. 3d) show a slight increase when the specific current increases from 0.1 to 5 A g$^{-1}$, suggesting the polarization effect of $MoS_{1.5}Te_{0.5}@C$ nanocables electrode is slighter than the others (Supplementary Fig. 17), which can be attributed to the following points: (1) the nanowire-in-nanotube structure in the $MoS_{1.5}Te_{0.5}@C$ nanocables not only function as 1D electron transport channels to improve the electrode conductivity but also can decrease the Na$^+$ diffusion distance and accelerate the Na$^+$ insertion/extraction rate, benefitting faster diffusion kinetics; (2) the carbon film surrounding $MoS_{1.5}Te_{0.5}$ nanosheets acts as a protective layer to facilitate the formation of a stable SEI, which can minimize charge transfer resistance; thus, the $MoS_{1.5}Te_{0.5}@C$ nanocables anode shows an improved rate performance. Moreover, the rate capability of the $MoS_{1.5}Te_{0.5}@C$ nanocable anode is better, especially at specific currents >1 A g$^{-1}$, than those reported in the literature for $MoS_2$ and $MoSe_2$ based materials for sodium-ion storage (Fig. 3e)[22,24,40–46]. Given that the $MoS_{1.5}Te_{0.5}@C$ nanocables display high rate capability, CV measurements at various scan rates are utilized to study the capacitive charge storage properties (Fig. 3f)[47,48]. The surface capacitive contribution ratio in $MoS_{1.5}Te_{0.5}@C$ nanocables becomes prominent with scanning rates from 0.2 to 1.4 mV s$^{-1}$ (Fig. 3g). Notably, the $MoS_{1.5}Te_{0.5}@C$ nanocables exhibits higher capacitive-controlled contribution than the other two samples (Supplementary Figs. 18 and 19). The electrochemical impedance spectroscopy (EIS) measurements of the $MoS_{1.5}Te_{0.5}@C$ nanocables and $CNT@MoS_2$ nanotubes active materials are disclosed (Supplementary Fig. 20 and Supplementary Table 4). One can observe that the $MoS_{1.5}Te_{0.5}@C$ nanocables shows the smaller semi-circle feature at high-frequency region and the higher slope of the tail in the low-frequency region than that of $CNT@MoS_2$ nanotubes, which are related to the charge-transfer resistance ($R_{ct}$) and the Na$^+$-ion diffusion resistance ($Z_w$), respectively, indicating the Te-doping is more favorable for electron transfer and mass transport. The $MoS_{1.5}Te_{0.5}@C$ nanocables reveal a good stability with a capacity of 355.4 mAh g$^{-1}$ after 500 cycles at 1 A g$^{-1}$, retaining 93.3% of the capacity retention, corresponding to a capacity fading rate of only 0.012% per cycle (Fig. 3h). On the contrary, both $MoS_{1.5}Te_{0.5}@C$ nanotubes and $CNT@MoS_2$ nanotubes electrodes show poorer cycle life, rate performance, and polarization.

Based on the Na-ion storage properties of the $MoS_{1.5}Te_{0.5}@C$ nanocables, we have assembled a dual-ion cell using the $MoS_{1.5}Te_{0.5}@C$-based electrode as anode, expanded graphite as cathode active material, and $NaPF_6$-based non-aqueous electrolyte solution. Figure 4a illustrates the charging–discharging mechanism of the dual-ion cell, where both electrodes and electrolyte play roles as active materials[49]. Upon the charging process, Na$^+$ cations migrate toward anode side and then undergo conversion reaction with $MoS_{1.5}Te_{0.5}@C$ nanocables electrode, while PF$_6^-$ anions simultaneously move to the cathode side and insert into the graphite layers of EG under the driving force of applied voltage. Subsequently, the discharge process is the inverse of the charge process, in which both charge carriers will be released from electrodes and integrated into electrolytes. The structural characterization and electrochemical behaviors of EG

cathode are also studied, as presented in Supplementary Fig. 21. The charge–discharge curve and the corresponding d$Q$/d$V$ differential curve exhibits multi-platform regions of the voltage profile (Fig. 4b and Supplementary Fig. 22), which signifies a typical profile of PF$_6^-$ intercalation/deintercalation into/from EG cathode in high potential and the sodiation/de-sodiation of $MoS_{1.5}Te_{0.5}@C$ anode in low potential. The CV curves of each electrode are also displayed in Supplementary Fig. 23, and the cut-off voltage of the dual-ion cells tested is selected between 1.5 and 4.7 V to avoid electrolyte decomposition and other side reactions (Supplementary Fig. 24). Besides, the anode/cathode mass ratio is controlled at about 1:4 to match the charge balance between anode and cathode to obtain the desired electrochemical energy storage performance (Supplementary Fig. 25).

The $NaPF_6$ concentration in electrolytes also exerts a pivotal role in affecting the performance of the sodium-based dual-ion cells, because the associated ions function as active charge carriers for cell operation. Hence, we study how the electrolyte concentration affects the electrochemical performance (Fig. 4c, d and Supplementary Fig. 26). Figure 4c exhibits the cycle performance of various $MoS_{1.5}Te_{0.5}@C$ nanocables∥EG dual-ion cells tested in various electrolyte formulations with concentrations of 1.0, 2.0, and 3.0 M at a specific current of 0.1 A g$^{-1}$. The capacity tends to increase with increasing the electrolyte concentration with delivering the highest capacity of 218.6 mAh g$^{-1}$ in 3.0 M with a high CE of 96.1%, which exceeds those of others of 1.0 M (101.3 mAh g$^{-1}$, 83.7%) and 2.0 M (141.4 mAh g$^{-1}$, 88.9%) after 100 cycles (the specific capacity is calculated based on the anode). In addition, the dual-ion cell with the 3.0 M electrolyte solution shows a high CE and short activation time compared to the other cells tested with different electrolyte concentrations (Supplementary Figs. 27 and 28). By contrast, the CE values in 1.0 and 2.0 M salt concentrations are less than 90%. According to the electrochemical performance of SDIBs in varied concentration electrolytes, it is proved that a high concentrated electrolyte can improve the electrochemical performance of SDIBs, which could be attributed to the stable SEI layer formation (Supplementary Fig. 29)[50].

Figures 4d, e presents the rate performance of sodium-based dual-ion cells at varied current rates (0.1–5 A g$^{-1}$) and the corresponding GCD curves, respectively. As shown in Fig. 4d, the capacities of the dual-ion cells were 214.2, 207.8, 195.9, 175.8, and 150.2 mAh g$^{-1}$ at 0.1, 0.2, 0.5, 1, and 2 A g$^{-1}$ in 3.0 M electrolyte, respectively. Even at a current rate of 5 A g$^{-1}$, the value can still be maintained as high as 100.9 mAh g$^{-1}$ and the capacity can also recover to 209.8 mAh g$^{-1}$ when the specific current went back from 5 to 0.1 A g$^{-1}$, which outperforms the other SDIBs employing 1.0 M or 2.0 M electrolyte. Figure 4e shows the GCD curves at the different current rates with slight charge/discharge plateau separation, revealing low electrode polarization. The fabricated sodium-based dual-ion cell shows an adequate long-term cycling lifespan at a high current rate of 1 A g$^{-1}$, as shown in Fig. 4f. The present SDIBs maintain a high specific capacity of 145.7 mAh g$^{-1}$ after 1500 cycles with only 0.0042% fading per cycle, which could be considered as relevant performance when compared to the current state of the art of SDIBs (Supplementary Table 5). We have also calculated the specific capacity values of the sodium-based dual-ion cells based on the total mass of the negative and positive electrodes. We have found that the sodium-based dual-ion cell cycled at a specific current of 20 mA g$^{-1}$ (based on the total mass of both electrodes) could still deliver a discharge capacity of 43.4 mAh g$^{-1}$, which is aligned with the state-of-the-art performance of SDIBs (Supplementary Fig. 30 and Supplementary Table 6). Meanwhile, the average discharge voltage of such sodium-based dual-ion cell still holds around 3.1 V after 1000 cycles (inset of Fig. 4f) and the "DIB" sign made

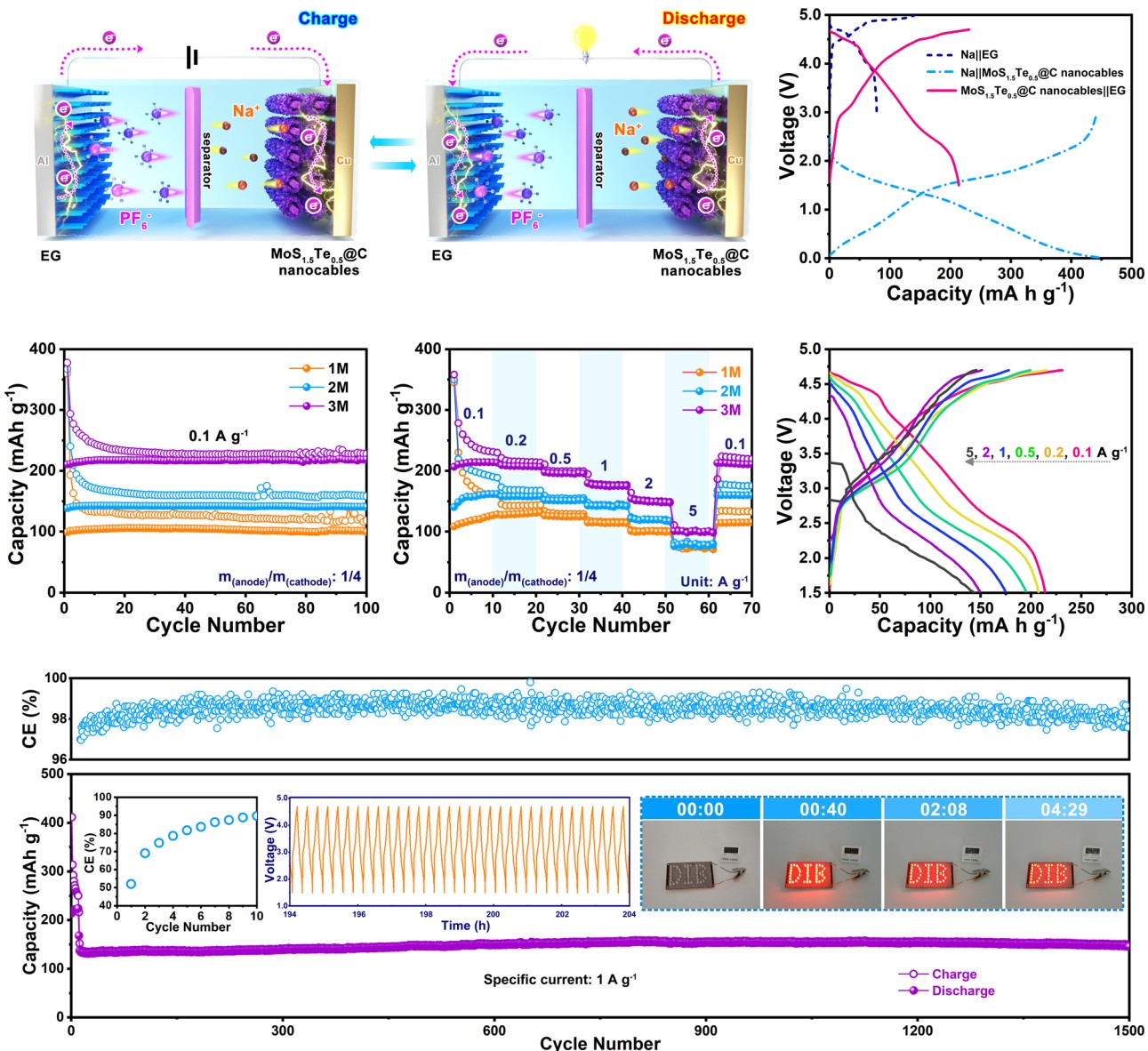

**Fig. 4 Electrochemical energy storage performance of the MoS$_{1.5}$Te$_{0.5}$@C nanocables||EG dual-ion cells. a** Schematic illustration of the MoS$_{1.5}$Te$_{0.5}$@C nanocables||EG dual-ion cell, **b** potential profiles for the Na metal||EG, Na metal||MoS$_{1.5}$Te$_{0.5}$@C nanocables and MoS$_{1.5}$Te$_{0.5}$@C nanocables||EG cells in 3 M NaPF$_6$ in EC/EMC/DMC (1:1:1, by volume) electrolyte solution at 25 °C. **c** Cycling stability at 0.1 A g$^{-1}$ and **d** rate performance profiles at various rates (from 0.1 to 5 A g$^{-1}$) for SDIBs with different NaPF$_6$ salt concentration (i.e., 1.0, 2.0, and 3.0 M) in EC/EMC/DMC (1:1:1, by volume). **e** Charging-discharging curves in a 3 M NaPF$_6$ in EC/EMC/DMC (1:1:1, by volume) electrolyte solution at 25 °C, and **f** long-term cycling stability of MoS$_{1.5}$Te$_{0.5}$@C nanocables||EG dual-ion cell at 1 A g$^{-1}$ in a 3 M NaPF$_6$ in EC/EMC/DMC (1:1:1, by volume) electrolyte solution at 25 °C. Inset in (**f**) the coulombic efficiency (CE) of initial 10 cycles (left), the corresponding charge-discharge curves at 1 A g$^{-1}$ from 192 to 206 h (middle) and the digital photographs of the "DIB" logo composing of 42 LEDs light up by one single cell (right).

of 42 light-emitting-diodes (LEDs) can be lighted up by one coin-cell for over 4 min (inset of Fig. 4f, Supplementary Fig. 31, and Movie 1), which further demonstrates the potential practical applications of this device. The cycled MoS$_{1.5}$Te$_{0.5}$@C nanocables electrode still holds an integrally tubular structure without marked pulverization after 1500 cycles at 1 A g$^{-1}$ as demonstrated by the ex situ postmortem electrode SEM and EDX measurements of Supplementary Fig. 32. This behavior suggests that the wire-in-tube structure can provide a larger buffer region for stress relief during the repeated sodiation/desodiation process. We also calculated the specific energy and power values of the sodium-based dual-ion cell (based on the total mass of the anode and cathode) which result to be of 171.4 and 305 W kg$^{-1}$,

respectively. These values are well-aligned with previously reported sodium-based dual-ion cells (Supplementary Fig. 33).

**Ex situ physicochemical characterizations.** To get a deep sight into the mechanistic details, the ex situ XRD, Raman, and TEM analyses were carried out to study the structural evolution of MoS$_{1.5}$Te$_{0.5}$@C nanocables anode and EG cathode under charge/discharge process, respectively. Figure 5a presents the charging-discharging curve of a MoS$_{1.5}$Te$_{0.5}$@C nanocables||EG dual-ion cell, during which the electrode at various stage-voltages marked by colored points was collected for the ex situ measurement. For MoS$_{1.5}$Te$_{0.5}$@C nanocables anode, the ex situ XRD patterns in

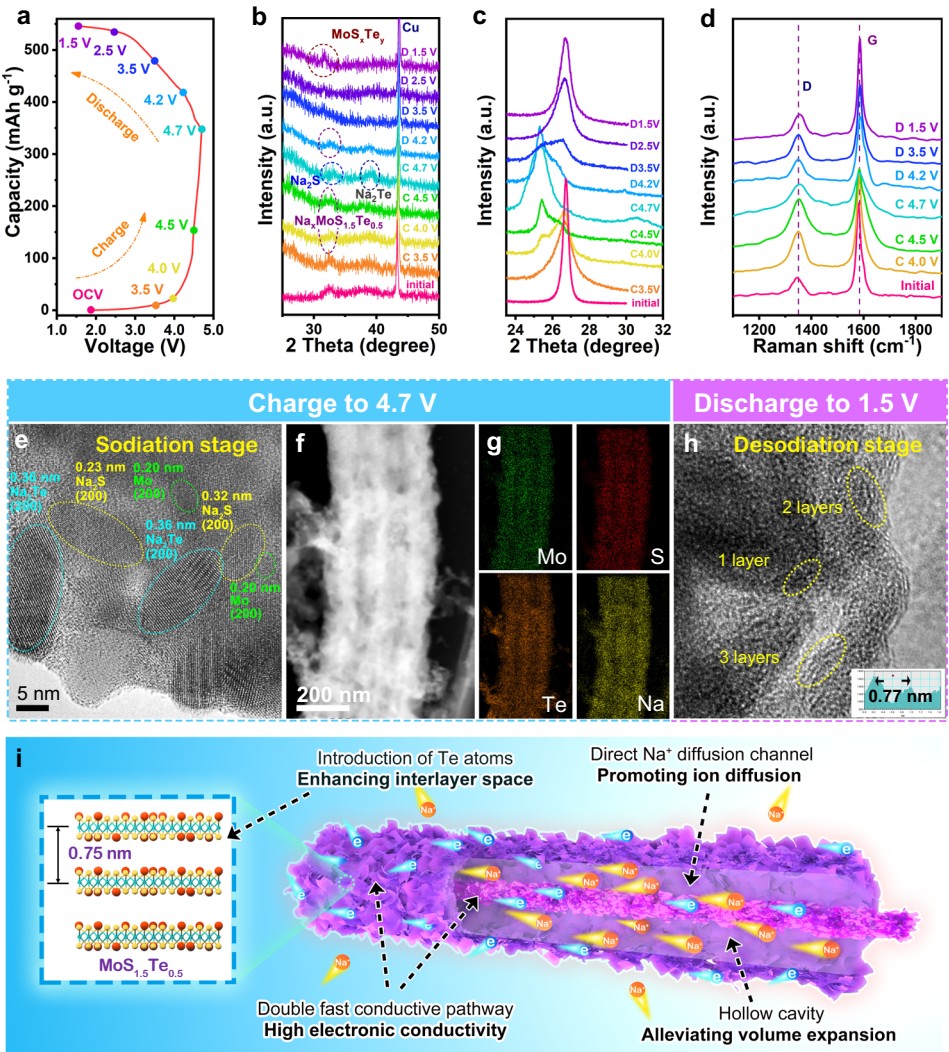

**Fig. 5 Structural evolution of MoS$_{1.5}$Te$_{0.5}$@C nanocables anode and graphite cathode in the 1st cell charge/discharge cycle. a** Charge–discharge curves of MoS$_{1.5}$Te$_{0.5}$@C nanocables||EG dual-ion cell marked by colorful dots in the first cycle. Ex situ XRD patterns of the MoS$_{1.5}$Te$_{0.5}$@C nanocables anode (**b**) and EG cathode (**c**). **d** Ex situ Raman spectroscopy of EG cathode. **e** HRTEM image, **f** STEM image, **g** elemental mappings at fully charging state, and **h** HRTEM image at fully discharging state for MoS$_{1.5}$Te$_{0.5}$@C nanocables anode. The ex situ postmortem measurements were carried out on electrodes cycled using the 3 M NaPF$_6$ in EC/EMC/DMC (1:1:1, by volume) electrolyte solution at a specific current of 0.1 A g$^{-1}$ and 25 °C. **i** Schematic illustration of the Na$^+$ storage mechanism and electronic conductivity in the MoS$_{1.5}$Te$_{0.5}$@C nanocables with stable "nanowire-in-nanotube structure".

Fig. 5b demonstrate the structural reversibility during the charging-discharging process. Before charging, at the position of open-circuit voltage (OCV), one reflection at 32.5° could be attributed to (100) crystal planes of MoS$_{1.5}$Te$_{0.5}$. With a continuous charging process, the reflection shifts to 32.9°, indicating the formation of Na$_x$MoS$_{1.5}$Te$_{0.5}$. When the voltage reaches 4.7 V, two reflections at 33.3° and 39.5° (marked by the circle) can be detected, which could be indexed to the crystals of Na$_2$S and Na$_2$Te, respectively, revealing the conversion evolution from Na$_x$MoS$_{1.5}$Te$_{0.5}$ into Na$_2$S and Na$_2$Te[51–53]. Besides, one can see one reflection at 31.5° assigning to the (100) crystal plane of MoS$_x$Te$_y$ at the end of the discharging process, suggesting the process is reversible. Importantly, the reflection (100) has slightly shifted to a lower angle than that of the pristine one, which could be accountable for an increased interlayer spacing for the Na$^+$ inserting and releasing process.

From the ex situ XRD patterns of the EG electrode in Fig. 5c, we can observe an intense graphitic (002) reflection at 26.5° in the initial stage. With the continuous charging process, the (002) reflection tends to weaken and shift negatively to a low angle,

while a new reflection at ~25.5° emerges and tends to intensify, indicating the expansion of the graphite layer with intercalation of anions (PF$_6^-$) into graphite. With the voltage charging up to 4.7 V, the reflections further split to 25.3° and 30.5°, demonstrating a form of stage-3 graphite intercalation compounds (Supplementary Table 7). During the discharge process, these reflections gradually became weak and finally disappear, whereas the intensity of (002) reflection successively increased, and both of them finally merge into one peak at the original diffraction position when discharged to 1.5 V, indicating the reversible de/intercalation of PF$_6^-$ anions under EG electrode[54]. On the other hand, the ex situ Raman results, based on the relative intensity between G peak (near 1580 cm$^{-1}$) and D peak (near 1350 cm$^{-1}$), also verify the reversible process during the charging–discharging process (Fig. 5d)[8].

The ex situ HRTEM measurements further verify the phase-evolution results of MoS$_{1.5}$Te$_{0.5}$@C nanocables anode. At the full sodiation stage in Fig. 5e, the layered MoS$_{1.5}$Te$_{0.5}$ nanosheets completely disappeared, and one can clearly observe several nanocrystal areas, relating to Na$_2$S, Na$_2$Te, and Mo

nanoparticles[51,55]. The EDS mapping diagrams exhibit the elements of Mo, S, Te, and Na are uniformly distributed over the wire-in-tube structural composites (Fig. 5f, g), suggesting that Na ions could be reversibly stored in the $MoS_{1.5}Te_{0.5}$@C nanocables electrode. When the cell discharges to 1.5 V, a layer of structured nanosheets can be observed again, demonstrating the reversibility of the conversion reaction (Fig. 5h)[56].

Based on the above analysis, the $MoS_{1.5}Te_{0.5}$@C nanocables electrodes present desirable electrochemical performance for $Na^+$ storage, due to its unique structural characteristics (Fig. 5i), including: (1) the nanowire-in-nanotube structure not only provides more electrode/electrolyte interface for sufficient charge transfer reaction but also builds a buffer region for volume expansion under continuous charge–discharge processes; (2) the substitution doping of Te atoms could narrow interlayer energy band of $MoS_2$, which enhances the electronic conductivity of electrode; (3) an enlarged layer spacing induced by large-sized Te atom introduction could also facilitate $Na^+$ migration by decreasing the Na-ion diffusion energy barrier. (4) Weak Mo–Te interaction is easier to break than that of Mo-S bands, leading to improved reversibility.

**Theoretical calculation and analyses**. The density functional theory (DFT) calculations are performed to gain a deeper insight into the anion defect properties and $Na^+$ storage behavior of electrode[57]. Supplementary Figs. 34 and 35 show the optimized geometry configurations of the $MoS_2$, $MoS_{1.5}Te_{0.5}$, and $MoS_{1.5}Te_{0.5}$ with defects and the corresponding $Na^+$ adsorbed structures. As shown in Fig. 6a, it should be noted that the Mo-Te bond distance in $MoS_{1.5}Te_{0.5}$ phase (0.27 nm) is significantly longer than the Mo-S bond (0.24 nm) in $MoS_2$, which makes the former easier to break during the conversion reaction, hence presenting faster reaction kinetics and higher reversibility. Figure 6b shows the crystal orbital Hamilton population (COHP) analyses based on the interaction of Mo-S bond in $MoS_2$ and $MoS_{1.5}Te_{0.5}$. We found the integrated COHP (ICHOP) of $MoS_2$ is smaller than that of $MoS_{1.5}Te_{0.5}$ around the Fermi level, which implies the introduction of Te would be conducive to weakening the Mo–S bond of $MoS_{1.5}Te_{0.5}$ and facilitating the bond breaking. As indicated in Fig. 6c, the defect-rich $MoS_{1.5}Te_{0.5}$ models reveal more negative values of the adsorption energy ($\Delta E_a$) than that of defect-free $MoS_2$ and $MoS_{1.5}Te_{0.5}$ models, which indicates that the abundant anion defects endow electrode with stronger adsorption for $Na^+$ [58]. Subsequently, we study the $Na^+$ diffusion behavior in the interlayers of $MoS_2$ and $MoS_{1.5}Te_{0.5}$ at different diffusion states (Fig. 6d, e). It can be noted that the $MoS_{1.5}Te_{0.5}$ model exhibits a lower value of 0.40 eV at the highest diffusion energy barrier site, whereas the corresponding value for $MoS_2$ is 0.58 eV, which indicates a much easier diffusion behavior of $Na^+$ on the $MoS_{1.5}Te_{0.5}$ interlamination (Fig. 6f). Figure 6g shows electron density differences models for studying the binding properties of the adsorbed Na atoms in all samples. One can see the defect-rich structures exhibit stronger electronic deficiency characteristics compared to the defect-free structures, suggesting the introduction of defects is more energetically benefited to capture $Na^+$, which would favor the higher redox reaction activity. The density of states (DOS) of $MoS_2$, defect-free $MoS_{1.5}Te_{0.5}$ and defect-rich $MoS_{1.5}Te_{0.5}$ are presented in Fig. 6h. There is an obvious bandgap in the bare $MoS_2$ model in the vicinity of the Fermi level but disappears in $MoS_{1.5}Te_{0.5}$ systems. This result demonstrates the DOS around the Fermi level is strengthened owing to Te-doping, which not only improves the electronic conductivity, but also increases the unpaired electrons near the Fermi level, thus leading to an increase in the reactivity of the material, and strengthening the

Na adsorption capacity. Thus, both the ionic and electronic transport (i.e., the electrical transport) would be accelerated by Te-doping. To understand the reaction process, the Gibbs free energy evolution from the initial phase to the final phase transformation reaction is simulated (Fig. 6i). Obviously, the reaction step of each electrode is downhill in free energy for transformation reaction, but it presents a more negative trend for the $MoS_{1.5}Te_{0.5}$. It means that the Na-ion storage on the $MoS_{1.5}Te_{0.5}$ electrode is more favorable during the electrochemical reaction process due to the lower free energy, thus resulting in efficient reaction kinetics, which is in accord with the experiment results. In addition, the evolution of atomic structure confirms the sodiation process involves the intercalation and conversion reaction mechanisms, which basically matches with ex situ XRD and HRTEM results (Supplementary Fig. 36). These results further demonstrate that the substitutional-doped Te in the $MoS_{1.5}Te_{0.5}$ system would greatly increase the electronic conductivity, decrease the $Na^+$ diffusion barrier, and reinforce the conversion reaction kinetics for $Na^+$, which improve the electrochemical performance of SDIBs.

## Discussion

In summary, we have developed a source-template strategy for the fabrication of a unique nanowires-in-nanotube structured $MoS_{1.5}Te_{0.5}$@C material. The favorable merits, including the unique nanowire-in-nanotube structure and physiochemical properties arising from Te doping, which guarantee to the $MoS_{1.5}Te_{0.5}$@C nanocables electrode improved electrochemical properties as anode for SDIBs when paired with EG cathode. Through optimizing concentration electrolyte and voltage window, the sodium-based dual-ion cells show appealing electrochemical energy storage performances even at high specific currents (i.e., >1 A g$^{-1}$).

## Methods

**Chemicals**. Tellurium oxide (99.99%), polyvinylpyrrolidone (PVP, K29-32, average Mw = 58 k) were purchased from Aladdin. $Na_2MoO_4 \cdot 2H_2O$ (AR), D-glucose (AR), anhydrous ethanol (99.5%), thiourea (AR) were purchased from the Sinopharm Chemical Reagent Co., Ltd. Na metal blocks (7440-23-5, purity ≥ 99.5%) were purchased from the Sinopharm Chemical Reagent Co., Ltd. The Na metal foils as the counter/reference electrode were punched into circular sheets with a diameter of 12 mm and thickness of 0.5 mm from the sodium blocks. The electrolyte components (purity of $NaPF_6$, EC, EMC, and DMC ≥ 99.9, 99.95, 99.98, and 99.98%) were provided by Suzhou DoDochem Ltd. The $H_2O$ content of the $NaPF_6$-based electrolyte solutions is less than 20 ppm. The Cu (thickness: ~9 μm, purity ≥ 99.8), Al foil (thickness: ~15 μm, purity ≥ 99.7), expanded graphite (C% content ≥ 96), and acetylene black (C% content ≥ 99.5) were provided by Shenzhen Kejing Star Technology Company. The Cu and Al foil as the current collector were punched into circular sheets with a diameter of 12 mm, respectively. The glass fiber separator (Whatman GF/D, thickness: ~675 μm, diameter: 16 mm, porosity: 2.7 μm) was purchased from GE Healthcare UK Limited. The above all the chemicals were used directly as purchased without further purification treatment.

**Synthesis of tellurium precursor**. In this synthesis of Te nanotubes, 1.44 g of tellurium oxide (9 mmol) and 1.8 g of polyvinylpyrrolidone were mixed in 150 mL ethylene glycol in the 250 mL round flask under continuous magnetic agitation and then heated to 200 °C within 20 min to get a clear solution. Subsequently, 0.9 g sodium hydroxide was added quickly into the above solution, while the transparent solution quickly turned dark gray. After 30 min of reaction at 200 °C, the obtained-sample were collected by filtration and washed with ethanol and distilled water several times. The procedure for synthesizing Te nanorods is similar to that of Te nanotubes except that the temperature is adjusted to 170 °C. The CNT was obtained after pyrolysis of PDA-coated Te nanotubes under a mixed Ar/H$_2$ (Ar/ H$_2$ = 90:10) at 800 °C for 2 h with a ramping rate of 3 °C min$^{-1}$.

**Synthesis of nanowire-in-nanotube structured $MoS_{1.5}Te_{0.5}$@C nanocables**. Typically, the 50 mg obtained-Te nanotubes and 100 mg D-glucose were dissolved in 30 mL of distilled water. After stirring for about 10 min, 150 mg of $Na_2MoO_4 \cdot 2H_2O$ and 300 mg of thiourea were added to the above solution, then sealed in a Teflon-lined stainless-steel autoclave (50 mL) with rotor and hydrothermally treated at 180 °C for 10 h in oil bath. After cooling to room temperature, the black product was collected by centrifugation, washed with ethanol, and

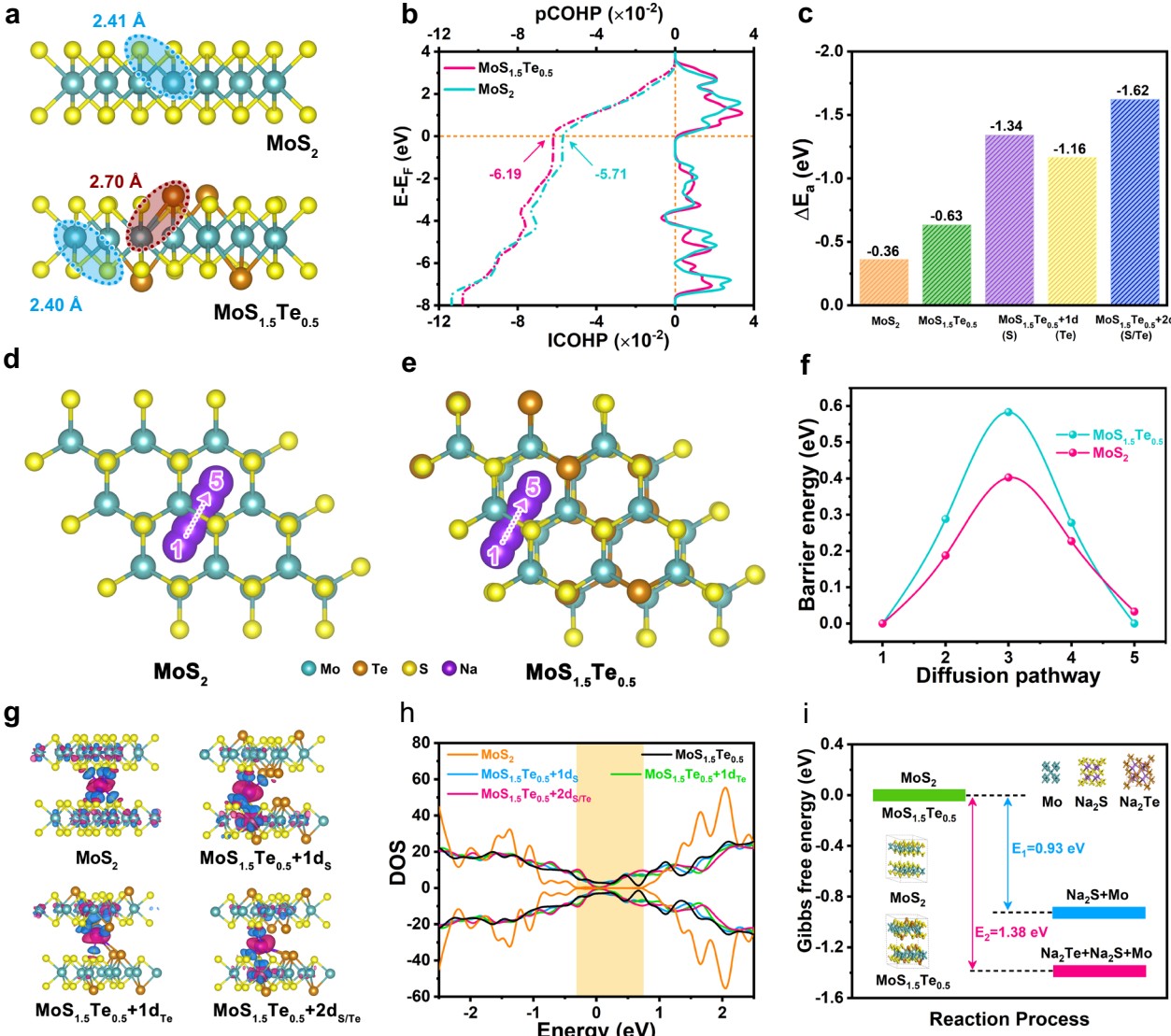

**Fig. 6 Theoretical calculation and analyses for the Te doped MoS₂. a** The optimized structures of MoS$_2$ and MoS$_{1.5}$Te$_{0.5}$. **b** Average pCOHP and the corresponding integral patterns for MoS$_2$ and MoS$_{1.5}$Te$_{0.5}$. The pCOHP is the projected crystal orbital Hamilton population. The ICOHP is the integrated COHP. **c** The adsorption energy ($\Delta E_a$) for sodium storage in all calculated models of anodes. **d, e** Schematic illustration for the diffusion path of a Na atom in the MoS$_{1.5}$Te$_{0.5}$ and MoS$_2$, and **f** corresponding diffusion energy barrier curves. **g** Electron density differences of intercalated Na in MoS$_2$, MoS$_{1.5}$Te$_{0.5}$, MoS$_{1.5}$Te$_{0.5}$ with one S defect, MoS$_{1.5}$Te$_{0.5}$ with one Te defect, MoS$_{1.5}$Te$_{0.5}$ with one S defect and one Te defect structure. **h** Total density of states (DOS) of MoS$_2$, MoS$_{1.5}$Te$_{0.5}$, MoS$_{1.5}$Te$_{0.5}$ with defects. **i** Free energy diagrams for the intercalation reaction and conversion reaction of MoS$_2$ and MoS$_{1.5}$Te$_{0.5}$.

distilled water several times and followed by vacuum drying for one night to obtain Te@C/MoS₂ composites. Finally, the above-obtained product was calcined at 800 °C under a mixed Ar/H₂ (Ar/H₂ = 90:10) atmosphere for 150 min with a heating rate of 3 °C min⁻¹ to obtain the nanowire-in-nanotube structured MoS$_{1.5}$Te$_{0.5}$@C nanocables. For composition, the products were synthesized with Te nanorods or CNT as precursors under the same conditions and named MoS$_{1.5}$Te$_{0.5}$@C nanotubes and CNT@MoS₂ nanotubes.

**Synthesis of nanowire-in-nanotube structured WS$_x$Te$_y$@C nanocables**. The procedure for the preparation of WS$_x$Te$_y$@C Nanocables is similar to that described above except that the Na₂MoO₄·2H₂O was replaced by Na₂WO₄·2H₂O.

**Synthesis of nanowire-in-nanotube structured ReS$_x$Te$_y$@C nanocables**. The procedure for the preparation of ReS$_x$Te$_y$@C Nanocables is similar to that described above except that the Na₂MoO₄·2H₂O was replaced by NH₄ReO₄.

**Characterizations**. The XRD patterns of the prepared samples were obtained on a Miniflex600 powder X-ray diffractometer using Cu Kα radiation in the 2θ range from 10° to 80° at a scan rate of 0.05° s⁻¹. The morphologies of the samples were

characterized by FESEM (Hitachi SU-8020), TEM, and HRTEM (Tecnai F20). Energy-dispersive X-ray spectroscopy analysis was carried out in the TEM. Nitrogen adsorption−desorption isotherms and BET surface area measurements were performed with an automated gas sorption analyzer (Hiden IGA100B). The measurements were performed at 77 K, and the pore-size distribution was derived using the DFT model. TGA of the sample was recorded with a thermogravimetric analyzer (Netzsch STA449F3) in air with a heating rate of 10 °C min⁻¹ from room temperature to 800 °C. The Raman spectrum was recorded on a Renishaw in Via Raman microscope (532 nm). XPS (ESCALAB 250) was used to determine the components on the surface of the samples. The inductively coupled plasma atomic emission spectroscopy measurements were tested on an Optima 7300 DV. The elemental analysis (EA) measurements were tested on EA mode (Vario EL Cube). The defect structure was measured by EPR spectra (Bruker-E500).

**Anode and cathode preparation**. The anode was fabricated by mixing active materials (80 wt%), acetylene black (10 wt%), and sodium carboxymethyl cellulose (10 wt%) into a uniform slurry, and followed to coated onto a copper foil. The pellets were dried in a vacuum at 80 °C for 24 h. The mass loading of the anode active material on each electrode (diameter: 12 mm) was ~1.2 mg cm⁻². The EG cathode was also prepared by a slurry coating procedure with the slurry consisting

of 80 wt% expanded graphite, 10 wt% acetylene black, and 10 wt% Poly(vinylidene fluoride) dissolved in N-Methyl pyrrolidone, and then coated on Al foil. The coated Al foils were also dried in a vacuum at 80 °C for 24 h. The loading mass of the cathode material (diameter: 12 mm) was ~5 mg cm$^{-2}$. The Na metal foil was punched into circular sheets with a diameter of 12 mm and a thickness of 0.5 mm from the sodium blocks. The glass fiber was punched into circular sheets with a diameter of 16 mm. cells.

**Electrochemical measurements**. The electrochemical measurements were investigated with 2032-type coin cells assembled in a glove box filled with argon atmosphere (H$_2$O < 0.5 ppm, O$_2$ < 0.5 ppm). For SIBs, the Na metal foils as the counter/reference electrode, 1 M NaPF$_6$ dissolved in ethylene carbonate/ethyl methyl carbonate/dimethyl carbonate (EC/EMC/DMC, 1:1:1, v/v/v) as electrolyte and glass fiber as the separator. For SDIBs, the as-prepared MoS$_{1.5}$Te$_{0.5}$@C nanocables samples, EG, and glass fiber were used as anode, cathode, and separator, respectively. The electrolyte with composition of 1.0, 2.0 and 3.0 M NaPF$_6$ in ethylene carbonate/ethyl methyl carbonate/dimethyl carbonate (EC/EMC/DMC, 1:1:1, v/v/v, H$_2$O < 10 ppm) was used in dual-ion cells. The active mass loading ratio of the anode (MoS$_{1.5}$Te$_{0.5}$@C cables) and cathode (EG) was approximatively 1:2, 1:4, and 1:6, respectively. The electrolyte/anode ratio in each cell was ~100 μL mg$^{-1}$. Then, the assembled cells were aged for 20 h to ensure that the system was fully wetted. The cell cycling experiments are carried out in a climatic chamber with a constant temperature of 25 °C. The electrochemical performances of the batteries were tested on a Land battery testing system (LANDTE Co., China) with the potential window of 0.01–3 V for SIBs and 1.5–4.7 V for SDIBs, and CV tests were measured on a CHI604E electrochemical station (CHI Instrument Co., Shanghai, China), respectively. For the long-term cycling performance at high specific currents, the SDIBs were first activated at 0.1 A g$^{-1}$ for ten cycles and then were operated at a higher specific current for a long cycle test. An ac voltage amplitude of 5.0 mV was employed to measure EIS within the frequency range from 10 mHz to 100 kHz. The specific capacity and specific currents were calculated based on the weight of anode materials.

**Calculation of *b* value**. The b value is applied to evaluate the pseudocapacitive behavior of the electrode. According to the power-law relation between the sweep scan rate (v) and the peak current (i), the Eqs. (1 and 2) can be provided:

$$i = av^b \tag{1}$$

$$\log(i) = b\log(v) + \log(a) \tag{2}$$

in which the *b*-value of 0.5 or 1.0 indicates a fully diffusion-dominated or surface-capacitive process, respectively.

**Calculation of capacitive contribution**. Quantitatively, the capacitive-dominated contribution can be separated based on the current response (i) at a fixed voltage (v), according to the Eqs. (3 and 4)

$$i(V) = k_1 v + k_2 v^{1/2} \tag{3}$$

$$i(V)/v^{1/2} = k_1 v^{1/2} + k_2 \tag{4}$$

where $k_1$ and $k_2$ are adjustable parameters, the $k_1 v$ stands for capacitive-controlled process, and the $k_2 v^{1/2}$ represents ionic-diffusion controlled process.

**Ex situ characterization**. The electrode for ex situ characterization were composed of 90 wt% active materials and 10 wt% sodium carboxymethyl cellulose to exclude the effect of acetylene black. As for the electrode for ex situ XRD and Raman, the corresponding process was as follows: first, the cells at different discharging/charging states were carefully disassembled inside an argon-filled glove box; then, the electrodes were washed with DMC three time and dried in the glove box to evaporate solvents; finally, the dry electrodes was sealed with Kapton tapes (~30 μm in thickness) for ex situ XRD and Raman characterization. For ex situ TEM measurement, the dried electrodes were immersed in ethyl alcohol solvents (≥99.8%) in a glass vessel and sealed with sealing film (Parafilm, PM996) inside an argon-filled glove box. Subsequently, the solutions were dispersed under intense ultrasonication for 10 min to dropwise add onto a copper grid, and then rapidly transferred into the TEM equipment.

**Calculation method**. DFT calculations were used with periodic super-cells under the generalized gradient approximation using the Perdew–Burke–Ernzerhof function for exchange-correlation. The interaction of ion-electron is described by a projected augmented wave. The Kohn–Sham orbitals were expanded in a plane-wave basis set with a kinetic energy cutoff of 30 Ry and the charge-density cutoff of 300 Ry. The Fermi-surface effects have been treated by the smearing technique of Methfessel and Paxton, using a smearing parameter of 0.02 Ry. The Brillouin-zones were sampled with a *k*-point mesh of 2 × 2 × 1. The calculation model of the MoS$_2$ and MoS$_{1.5}$Te$_{0.5}$ were constructed with 3 × 3 lateral periodicities shown in Supplementary Fig. 34. The vacuum layer was ~1.5 nm to decouple adjacent atomic slabs in the *z*-direction. The nudged elastic band (NEB) method was used to evaluate the transition state and activation energy for one Na atom migrating from one hollow site in MoS$_2$ and MoS$_{1.5}$Te$_{0.5}$ to another hollow site. To analyze the sodiation process of the MoS$_{1.5}$Te$_{0.5}$, the lowest energy structures including different sodium content are reoptimized using the vc-relax method. All the DFT calculations are implemented by the PW and NEB modules contained in the Quantum ESPRESSO distribution. Considering the reaction from Mo$_z$S$_x$Te$_y$ to Na$_2$S, Na$_2$Te, and Mo according to the following Eq. (5)

$$\text{Mo}_z\text{S}_x\text{Te}_y + 2(x+y)\text{Na} \rightarrow z\text{Mo} + x\text{Na}_2\text{S} + y\text{Na}_2\text{Te} \tag{5}$$

$\Delta G$ can be approximated by the total energies from DFT calculations neglecting the entropic contributions (0 K), according to Eq. (6)[59].

$$\Delta G = [xE_{(\text{Na}_2\text{S})} + yE_{(\text{Na}_2\text{Te})} + zE_{(\text{Mo})} - E_{(\text{Mo}_z\text{S}_x\text{Te}_y)} - 2(x+y)E_{(\text{Na})}]/2(x+y) \tag{6}$$

where $E_{(\text{Na2S})}$, $E_{(\text{Na2Te})}$, and $E_{(\text{MozSxTey})}$ are the DFT total energies at the optimized state, $E_{(\text{Na})}$ and $E_{(\text{Mo})}$ is the energy per atom in the respective metal phase.

**Calculation of the specific energy and power (based on the total mass of both anode and cathode materials)**. The cell-level specific energy E and specific power P are calculated according to the following Eqs. (7 and 8):

$$E = \int_{t_1}^{t_2} V I dt = \frac{V_{\max} + V_{\min}}{2} \times It \times \frac{1}{3600} \tag{7}$$

$$P = \frac{3600 \times E}{t} \tag{8}$$

where *t* (s) is the discharge time, *I* (A g$^{-1}$) is charge/discharge current, $V_{\max}$ (V) is the discharge potential excluding the IR drop and $V_{\min}$ (V) is the potential at the end of discharge voltages, *E* is the specific energy (Wh kg$^{-1}$), and *P* is the specific power (W kg$^{-1}$).

**Reporting summary**. Further information on research design is available in the Nature Research Reporting Summary linked to this article.

## Data availability

The data that support the plots within this paper and another finding of this study are available from the corresponding author upon reasonable request.

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

## Acknowledgements

We appreciate support from the National Natural Science Foundation of China (Project Nos. 21875253, 51872048, and 52073061), Natural Science Foundation of Fujian Province, China (Nos. 2021J01430167 and 2021J02020), CAS-Commonwealth Scientific and Industrial Research Organization (CSIRO) Joint Research Projects (121835KYSB20200039), and Scientific Research and Equipment Development Project of CAS (YJKYYQ20190007).

## Author contributions

Y.L. and Z.W. conceived the idea and co-wrote the paper; Y.L. carried out the main experiments, including experimental planning, physical characterization, electrochemical measurements, and designed all the graphics outlined in the work. X.H. interpreted the electrochemical measurement results and data analysis. J.L. carried out the DFT calculation; G.Z. carried out the BET characterization and designed the LEDs of "DIB" logo; J.Y. did the SEM measurements; H.Z., Y.T., and Z.W. supervised the research. All authors discussed the results and contributed to writing the paper.

## Competing interests

The authors declare no competing interests.
