## [Peer Review File · Nature Communications]

Reviewers' comments:

Reviewer #1 (Remarks to the Author):

The paper reported nanowire-in-nanotube MoS_{1.5}Te_{0.5}@C nanocables and applied them as an anode material to store Na⁺. Despite the nice morphology of the material, the work was carried out using mainstream approaches and I do not find a sufficient level of scientific novelty and insights that could benefit the energy field. The idea that the authors pitched in the introduction is not convincing and reads generic. I recommend a rejection of the paper.

1. Anion substitution has been widely studied in Na- and K-ion batteries, particularly for 2D metal chalcogenides, e.g., MoS₂, due to the ease of the substitution. Anionic defects have been often observed due to the size difference between pristine and doping anions, for which there wasn't solid data in the paper as the ERS signal could be caused by the defects carried by MoS₂ itself.
2. The bottleneck of dual-ion batteries lies in the limited capacity of carbon cathodes, as seen from this paper that only around 100 mAh g⁻¹ was obtained even expanded graphite sits close to the upper end of capacity. I don't see sufficient justifications of developing large-capacity anodes to overcome the bottleneck, due to the fact that a high cathode/anode mass ratio inevitably increases the total mass of the cell, which cannot guarantee an increase of the energy density of the cell. Previous study has shown that energy density of full cells starts declining once cathode/anode mass ratio exceeds a certain level. Also, the discharge/charge voltages of Mo(S,Te) are not ideal for an anode of dual-ion batteries and they are pushing the voltage limit of carbon cathodes further up, which should be avoided in practical applications. In addition, the authors pitched the high conductivity of Te over S. Since Te is not formed during the Na storage process, why is the high conductivity of Te even relevant?

Additionally, there are other issues the authors need to address before submitting the paper somewhere else.

1. Solid experimental proof of anionic defects, perhaps atomic-resolution TEM?
2. The correlation between surface area and initial CE: the difference of the surface areas between the samples is not significant enough to cause the discrepancy of the observed initial CEs.
3. Capacity of MoS_{1.5}Te_{0.5}: have the authors applied the change of the molecular weight caused by the doping to the calculation of capacity? If not, a capacity decrease is expected compared to the reported values.
4. Simulations: it has become common that simulations like the one presented in this paper are used to justify conversion-type electrode materials. This is not really accurate. Major crystal structure rebuilding occurs and metal forms during discharging, and these changes are not taken into consideration of the simulations. Also, there is no evidence of the presence of anionic defects and how the defects change after a conversion reaction, which makes the simulation results less convincing.

Reviewer #2 (Remarks to the Author):

This manuscript reports nanowire-in-nanotube structured MoS_{1.5}Te_{0.5}@C nanocables synthesized by a source-template synthetic strategy. The composite has a one-dimensional nanowire-in-nanotube architecture with a highly porous structure. As a result, it demonstrates excellent electrochemical performance as an anode material in sodium based dual-ion batteries. The materials and electrochemical performance are systematically investigated using multiple characterization and theoretical techniques. However, there are still some questions needing to be clarified.

1. Although excellent performance is presented for the nanowire-in-nanotube structure MoS_{1.5}Te_{0.5}@C nanocables, it is hard to understand why the MoS_{1.5}Te_{0.5}@C nanotubes have an inferior performance. They both have one-dimensional and porous structure as well as similar chemical composition. There is no intrinsic difference in morphology and structure. Actually, the MoS_{1.5}Te_{0.5}@C nanotubes should have advantages over nanocables in terms of volume

accommodation. This manuscript needs to clarify this point. Why do we need nanocables but not nanotubes?

2. A large amount of S atoms are substituted by Te atoms (25%). This should have significantly effects on the crystalline structure and chemical properties. How are Te atoms incorporated into the MoS₂ crystalline frameworks? More characterization should be conducted to clearly understand the doping effects.

3. In the main text, figures are not correctly cited. For example, the discussion on electrochemical performance cites Figure 1, but it should be Figure 2. "Figure 4b presents the charging-discharging curve of MoS_{1.5}Te_{0.5}@C nanocables//EG SDIBs" should be "Figure 4d". Figure 4d and Figure 5c appear before Figure 4a and Figure 5a. This is weird and unlogical. Please check.

4. Figure 3b shows significant effects of electrolyte concentration on cycling performance. But no explanation on this. What's the reason?

5. Figure 3f, the show up on the lighting diodes doesn't make sense to this work, and can be removed or moved to SI.

AUTHOR'S RESPONSE TO REVIEWERS

Referee #1

General Comment: *The paper reported nanowire-in-nanotube MoS_{1.5}Te_{0.5}@C nanocables and applied them as an anode material to store Na⁺. Despite the nice morphology of the material, the work was carried out using mainstream approaches and I do not find a sufficient level of scientific novelty and insights that could benefit the energy field. The idea that the authors pitched in the introduction is not convincing and reads generic. I recommend a rejection of the paper.*

Response: We thank the reviewer for his/her critical comments and valuable suggestion that brought up excellent points for further improving the quality of this submission. We got known that the reviewer holds the concerns on the scientific novelty and the insights for the energy field. We would thus take this opportunity to illustrate the novelty and the positive effects for the fields of energy storage in this work. **Above all**, we reported, in this work, an innovative source-template route (Te nanotube as both template and Te-doping source) to synthesis of the nanowire-in-nanotube MoS_{1.5}Te_{0.5}@C nanocables. In this regard, the present method stands in stark contrast to the previous method for the synthesis of hollow nanotubes that normally require the hard template, which suffer from relatively high manufacturing costs, complicated procedures, and the use of hazardous acids/alkalis for the template removal. In this context, this work shows novelties in terms of nanostructure design and materials synthesis. **Second**, the distinctive properties in MoS_{1.5}Te_{0.5}@C nanocables, including expanded interlayers distance, enhanced electronic conductivity and suitable buffer space, are greatly favorable toward sodium ions storage with fast kinetics, high reversibility, and robust structural stability. In addition, the SDIBs assembled by paring MoS_{1.5}Te_{0.5}@C nanocables anode with graphite cathode was exemplified as one instance to verify its potential application for Na⁺ storage, which presents a long-term lifespan with 97% capacity retention over 1500 cycles as well as an outstanding rate capability. **Last but not at least**, we conducted an in-depth and systematic study on understanding the electrochemical mechanism by systematic electrochemical tests, ex situ XRD, Raman, TEM, kinetics analysis (CV experiments and EIS spectra analysis) and

density functional theory (DFT) calculations, which also presents insightful analysis and view on the electrochemical mechanism.

We have made a substantial revision by taking full consideration of her/his comments. Below please find our point-by-point responses to each comment.

Comments 1: *Anion substitution has been widely studied in Na- and K-ion batteries, particularly for 2D metal chalcogenides, e.g., MoS₂, due to the ease of the substitution. Anionic defects have been often observed due to the size difference between pristine and doping anions, for which there wasn't solid data in the paper as the ERS signal could be caused by the defects carried by MoS₂ itself.*

Response: We appreciate the reviewer for his/her kind suggestions and valuable comments. We agree the reviewer's opinion that substitution can easily occur on the 2D MoS₂. Actually, one can also observe substituting defects in the CNT@MoS₂ sample, as evidenced by a weak EPR signal (Figure R1). Nevertheless, the MoS_{1.5}Te_{0.5} samples almost showed one-order magnitude higher signal of intensity than that of CNT@MoS₂ thanks to the increased anionic defects by Te doping in MoS_{1.5}Te_{0.5}, it is thus rational that the introduction of Te atoms play a key role in increasing the defect concentration. (Please see page 8 in revised manuscript).

Figure R1. EPR results for the CNT@MoS₂ nanotubes (a) and MoS_{1.5}Te_{0.5}@C nanocables (b).

Comments 2: *The bottleneck of dual-ion batteries lies in the limited capacity of carbon*

cathodes, as seen from this paper that only around 100 mAh g⁻¹ was obtained even expanded graphite sits close to the upper end of capacity. I don't see sufficient justifications of developing large-capacity anodes to overcome the bottleneck, due to the fact that a high cathode/anode mass ratio inevitably increases the total mass of the cell, which cannot guarantee an increase of the energy density of the cell. Previous study has shown that energy density of full cells starts declining once cathode/anode mass ratio exceeds a certain level. Also, the discharge/charge voltages of Mo(S,Te) are not ideal for an anode of dual-ion batteries and they are pushing the voltage limit of carbon cathodes further up, which should be avoided in practical applications. In addition, the authors pitched the high conductivity of Te over S. Since Te is not formed during the Na storage process, why is the high conductivity of Te even relevant?

Response: The reviewer raised a quite important issue on how to improve the total capacity of dual-ion batteries due to the limited capacity of carbon cathodes. We are grateful to take this opportunity to clarify this point.

The recent progress in this field implies that the SDIBs face several critical challenges, including: i) the limited capacity of graphite cathode; ii) the continuous electrolyte decomposition at high work voltage (> 4.5 V); iii) the relatively low capacity and slow dynamics of Na⁺ in anode side. We thus agree the reviewer's views that the cathode is considered as one of the major bottlenecks for enhancing its performance due to limited capacity, yet there is still grand space for improving the anode materials by addressing issues of low capacity, slow kinetics, safety risks and short lifespan. That is why there have been a large amount of research works striving to exploring high-performance anode materials for the batteries (including DIBs) that has a limited capacity of cathode. **One typical example is that the silicon nanomaterial has been explored as anode of lithium dual-ion batteries and has received remarkable result.^[1]** In this context, the focus in this submission is to develop desirable anode material for Na⁺ storage, the as-prepared MoS_{1.5}Te_{0.5}@C nanocables indeed show attractive electrochemical properties for Na⁺ storage with a high capacity and rapid Na⁺ reaction kinetics. Sodium based dual-ion batteries was assembled by pairing MoS_{1.5}Te_{0.5}@C nanocables anode with a commercial and cheap graphite cathode to exemplify its potential application.

Ideally for a full cell, the capacities (C) released at the cathode and the anode should be well balanced (Eq. R1), so that the cathode and the anode can be fully used. Therefore, the specific capacity based on the total mass of anode and cathode for a full cell with a fixed cathode capacity can be rationally increased by **enhancing the mass loading of cathode (m_c)** or decreasing the mass loading of anode (m_a) (Eq. R2):

$$Q_{\text{cell}} = C_{\text{c-s}} \times m_c = C_{\text{a-s}} \times m_a \quad (\text{R1})$$

$$C_{\text{T-s}} = \frac{Q_{\text{cell}}}{m_c + m_a} = \frac{C_{\text{c-s}}}{1 + m_a/m_c} \quad (\text{R2})$$

(Where $C_{\text{a-s}}$, $C_{\text{c-s}}$, $C_{\text{T-s}}$, m_a , and m_c are the specific capacity (mAh g^{-1}) based on the mass (mg) of anode, cathode, and the mass of anode and cathode, respectively, Q_{cell} is the released capacity (mAh)).

As expected, the EG cathode in this work displays a capacity of 91 mAh g^{-1} being close to those of the previous work (*Adv. Energy Mater.* **2018**, 9, 1803260; *Small* **2017**, 13, 1702449). Generally, high cathode loading is essential for fabricating high energy-density cells toward practical full cells. The high-performance anode material of $\text{MoS}_{1.5}\text{Te}_{0.5}@\text{C}$ nanocables, enables us loading more graphite cathode in SDIBs full cell with a mass ratio of anode/cathode of 1:4, which can result in an improved energy density based on total mass, to a certain degree exhibiting advantages over the recent works of SDIBs. (Figure R2 and Table R1).

In addition, the energy and power densities of the $\text{MoS}_{1.5}\text{Te}_{0.5}@\text{C}$ nanocables//EG SDIBs were calculated based on the total mass of both anode and cathode materials using the equations below (R3, R4):

$$E = \int_{t_1}^{t_2} VI dt = \frac{V_{\text{max}} + V_{\text{min}}}{2} \times It \times \frac{1}{3600} \quad (\text{R3})$$

$$P = \frac{3600 \times E}{t} \quad (\text{R4})$$

where t (s) is the discharge time, I (A g^{-1}) is charge/discharge current, V_{max} (V) is the discharge potential excluding the IR drop and V_{min} (V) is the potential at the end of discharge voltages, E is the energy (Wh kg^{-1}) and P is the power densities (W kg^{-1}). (Please see page 7 in supporting information). As shown in Figure R3, the $\text{MoS}_{1.5}\text{Te}_{0.5}@\text{C}$ nanocables//EG SDIBs

deliver high energy densities of 192, 183.7, 171.4, 151, 125.5 and 70.1 Wh kg⁻¹ at the power densities of 61.9, 123.2, 305, 601, 1168, 2500 W kg⁻¹, respectively. (Please see page 36 in supporting information). Impressively, the energy density of 171.4 Wh kg⁻¹ for full cell under a powder density of 305 W kg⁻¹ is competitive to those of previously reported sodium-based devices and other type of energy storage devices, highlighting the promising potential in practicability.^[3-11] (Please see page 15 in revised manuscript).

We agree with the reviewers that a low discharge/charge voltage of anode conduce to improve energy density. The carbon materials exhibit a fairly low de/intercalation voltage, whereas its capacity is limited by the small number of crystallographic sites for storing sodium ions (~30 mAh g⁻¹ for NaC₆₄); Alloys with enhanced capacity have thus been investigated, but tend to undergo extra volume expansion during sodiation/de-sodiation with unsatisfactory cycling stability. In this work, the MoS_{1.5}Te_{0.5}@C nanocables operates at a relatively high voltage platform that can enhance the safety of full cell. Notably, the as-assembled SDIBs still deliver a high energy density thanks to the decently high voltage of dual-ions batteries, which outperforms previously reported sodium-based devices and other type of energy storage devices (Figure R3). In addition, the focus in this work is to explore and to develop an advanced Na⁺ storage anode material, which enriches the type of anode materials. We anticipate that such an artful approach can be generalized to a wider range of 1D dimensional configurations with desirable phase compositions, structural configuration, and properties.

We would thank the reviewer again for pointing out the improper description regarding “*the high conductivity of Te over S*”. We have accordingly revised the related descriptions in the revised manuscript: Actually, Tellurium (Te) element, as one member of chalcogens group, has a larger atomic radius (~1.43 Å) than sulfur (S, ~1.05 Å), which makes the valence electron of Te atom less restrained by the nuclear charge. Thus, the Te tend to amplify interlayer spacing and enhance the conductivity of MoS₂ when doped into MoS₂ by narrowing interlayer band gaps of MoS₂ with manifesting metallic properties, which can effectively boost electrochemical process dynamics.^[2] (Please see page 4 in revised manuscript).

Figure R2. The cycling performance at (a) 20 mA g^{-1} and (b) 200 mA g^{-1} and (c) the rate performance of the SDIBs (The specific capacity and current densities were calculated based on the total mass of the cathode and anode active materials). (The initial specific capacity of the SDIB is about 216.8 mAh g^{-1} based on the anode material. After including the cathode material mass, which is about 4 times of the anode material mass, the capacity calculated based on both electrode materials is about 43.4 mAh g^{-1} ($\approx 216.8 \text{ mAh g}^{-1}/(1 + 4)$)).

Figure R3. Energy and powder densities of our prepared SDIBs in comparison with those of previously reported sodium-based devices (a) and other energy storage devices (b) (the energy density and power density of SDIBs are based on the total mass of both anode and cathode materials). (Please see the Figure S31 in page 36 from the revised supporting information)

Table R1. Cycle performance comparison of MoS_{1.5}Te_{0.5}@C nanocables//EG SDIBs with various recently reported SDIBs. (The specific capacity and current densities were calculated based on the total mass of the cathode and anode active materials.)

Materials	Voltage range (V)	Current density (mA g ⁻¹)	Cycle number	Specific capacity (mAh g ⁻¹)	m _{anode} /m _{cathode} (mg)	Reference
MoS_{1.5}Te_{0.5}@C nanocables//EG Na-DIBs	1.5-4.7	20	100	43.4	1.0/4.0	This work
		200	1500	29.2		
		1000	Based on rate performance	20.2		
Soft carbon//graphite Na-DIBs	2.0-4.7	333	800	18	1.5/2.5-3.0	(12)
Hard carbon//graphite Na-DIBs	2.0-4.7	500	200	22.2	1.0/1.8	(15)
Coronene//Na₂Ti₃O₇ Na-DIBs	1.5-3.5	100	5000	19.3	0.9/4.5	(3)
P-doped hollow carbon//EG Na-DIBs	2.0-4.7	100	1500	24.2	1.0/4.0	(4)
N,S-doped MoS₂@C nanosheets//EG Na-DIBs	1.0-4.0	42.9	300	25.7	1.5/2.0	(13)
		428.5	5000	16.8		
P-doped soft carbon//graphite Na-DIBs	2.0-4.7	125	100	27.5	1.0/3.0	(14)
		250	900	20.2		

MoS₂/C-G//EG Na-DIBs	1.0-4.0	100	200	27.5	1.0/1.0	(16)
Hard carbon//graphite Na-DIBs	1.5-4.8	45	100	21.8	1.0/1.2	(17)

The other issues

Comments 1: *Solid experimental proof of anionic defects, perhaps atomic-resolution TEM?*

Response: We appreciate the reviewer for his/her kind suggestions. Accordingly, we conducted the atomic-resolution TEM tests and have provided the atomic-resolution STEM images for MoS_{1.5}Te_{0.5}@C nanocables (Figure R4) in the revision. The aberration-corrected high-angle annular dark-field scanning TEM (HAADF-STEM) image of MoS_{1.5}Te_{0.5}@C nanocables reveals that there are a few brighter spots occupying some anionic sites with distinguishable contrast, corresponding to the heavier and larger Te atoms. Besides, it can be observed that several anions have vanished in the selected regions (as marked by the white circles), implying the successful introduction of anionic defects.^[18] (Please see page 6 in the revised manuscript).

The possible formation mechanism of anionic vacancies is explained as follows: when the S atoms in MoS₂ crystal lattices are partially replaced by larger Te atoms, the interspace of S atoms in crystal lattices may be squeezed and S atoms may shift in the crystal lattices, leading to the formation of anionic vacancies in the crystal lattices.

Figure R4. The magnified high-resolution STEM images of $\text{MoS}_{1.5}\text{Te}_{0.5}@\text{C}$ nanocables with the intensity profile of the colored areas. (Please see the Figure S3 in page 10 from the revised supporting information)

Comments 2: *The correlation between surface area and initial CE: the difference of the surface areas between the samples is not significant enough to cause the discrepancy of the observed initial CEs.*

Response: The reviewer has raised a very good point and we appreciate that the reviewer noticed such a critical detail. There are many factors that affect the initial Coulombic efficiency (ICE) of electrode materials, such as the specific surface area, surface heteroatom content, and structure of electrodes and so on. We agree with the referee that the initial Coulombic efficiency (ICE) is not just affected by the specific surface area. In the present work, the ICE of $\text{MoS}_{1.5}\text{Te}_{0.5}@\text{C}$ nanocables is 83.2%, while this value is 73.9% for $\text{CNT}@\text{MoS}_2$ nanotubes. The $\text{MoS}_{1.5}\text{Te}_{0.5}@\text{C}$ nanocables possess the lower specific surface area and hold favorable structure and properties for improving ICE, including (1) the few-layers and expanded

interlayers distance of $\text{MoS}_{1.5}\text{Te}_{0.5}@C$ nanosheets shorten the Na^+ ion diffusion pathways; (2) Te doping tends to enhance the conductivity; (3) the robust carbon matrix protection could effectively avoid the direct contact between $\text{MoS}_{1.5}\text{Te}_{0.5}$ nanosheets and electrolyte to reduce the decomposition of electrolyte and the amount of irreversible SEI, thus resulting in high reversibility of the conversion reaction in $\text{MoS}_{1.5}\text{Te}_{0.5}@C$ nanocables hybrids and achieving high coulombic efficiency. Therefore, the $\text{MoS}_{1.5}\text{Te}_{0.5}@C$ nanocables exhibit considerably high ICE with high reversibility for Na^+ ions storage.

Comments 3: *Capacity of $\text{MoS}_{1.5}\text{Te}_{0.5}$: have the authors applied the change of the molecular weight caused by the doping to the calculation of capacity? If not, a capacity decrease is expected compared to the reported values.*

Response: Thanks for the reviewer's valuable comments. The calculation of capacity is based on total mass of active materials ($\text{MoS}_{1.5}\text{Te}_{0.5}$) instead of MoS_2 . We accordingly clarify this in the experimental section.

Comments 4: *Simulations: it has become common that simulations like the one presented in this paper are used to justify conversion-type electrode materials. This is not really accurate. Major crystal structure rebuilding occurs and metal forms during discharging, and these changes are not taken into consideration of the simulations. Also, there is no evidence of the presence of anionic defects and how the defects change after a conversion reaction, which makes the simulation results less convincing.*

Response: The reviewer has offered us with valuable comments with a clear goal to make the manuscript better; we thus would appreciate her/him very much. First of all, we agree with the referee that the DFT calculation of electron density differences and binding energy has been widely utilized in recent works and may not be really accurate especially for justifying the conversion-type electrode materials. In order to further simulate the electrochemical reaction, we have thus performed additional DFT calculations to investigate the phase transformation of $\text{MoS}_{1.5}\text{Te}_{0.5}$ through both equilibrium and nonequilibrium sodiation pathways.

As shown in Figure R5, several intermediate sodium-ion intercalated phases are predicted and identified according to DFT calculations. During the initial sodiation process, the anion defect sites present stronger electron deficiency characteristics, so that Na^+ tend to insert at anion defect sites, since the anion defects are more likely to gain electrons from Na atoms. The Na^+ intercalated $\text{MoS}_{1.5}\text{Te}_{0.5}$ lattice would form an intercalated structure when the number of sodium ions is less than or equal to the number of Mo ions. Above the thresholds, Mo ions start to be reduced and the conversion reaction will occur. The final phase after Na^+ insertion includes Mo, Na_2S and Na_2Te . (Please see page 41 in the supporting information). The evolution of atomic structure confirms the sodiation process involves the intercalation and conversion reaction mechanisms, which basically matches with *ex-situ* XRD and HRTEM results. (Please see page 19 in the revised manuscript).

In terms of the HAADF-STEM image (Figure R4), individual vanished atom sites marked by white circles are observed, revealing the existence of anion defects. In order to further study how anionic defects evolved after conversion reaction, we performed *ex-situ* electron paramagnetic resonance (EPR) tests basing on half-cell, which provide us with evidence to unveil the underlying mechanism. As shown in Figure R6, the dominant signatures of g-value at 2.003 can be observed when the cell discharged to 0.5V, which is assigned to the unpaired electrons trapped by sulfur vacancies, demonstrating the anionic defects still exist during the Na^+ intercalation process. The intensity of EPR signal becomes obviously weak at the final discharging state (0.01V), suggesting the formation of conversion products (Mo, Na_2S , Na_2Te). When the cell is further charged to 3.0 V, the EPR signal corresponding to sulfur vacancies displays an obvious increase, which may be probably attributed to rebuilding of MoS_xTe_y crystal structure. Therefore, the anionic defects tend to be reversible after the conversion reaction. These results are agreement with the *ex-situ* HRTEM and XRD analysis. It should be pointed out that the nanocables structure was well maintained during the charging-discharging processes, which may provide confined space for the conversion reaction and thus ensure the structure reversibility.

Figure R5. Calculated sodiation process of $\text{MoS}_{1.5}\text{Te}_{0.5}$ by DFT. In sodiation process, the corresponding intermediates are identified by DFT. (Please see the Figure S36 in page 41 from the revised supporting information)

Figure R6. a) Charge-discharge curves of $\text{MoS}_{1.5}\text{Te}_{0.5}@C$ nanocables//Na half-cell marked by colorful dots in 1st cycle, b) EPR spectra of $\text{MoS}_{1.5}\text{Te}_{0.5}@C$ nanocables electrode at different charging/discharging states.

Referee #2

This manuscript reports nanowire-in-nanotube structured $\text{MoS}_{1.5}\text{Te}_{0.5}@C$ nanocables synthesized by a source-template synthetic strategy. The composite has a one-dimensional nanowire-in-nanotube architecture with a highly porous structure. As a result, it demonstrates excellent electrochemical performance as an anode material in sodium based dual-ion batteries. The materials and electrochemical performance are systematically investigated using multiple characterization and theoretical techniques. However, there are still some questions needing to be clarified.

Comment 1: *Although excellent performance is presented for the nanowire-in-nanotube structure $\text{MoS}_{1.5}\text{Te}_{0.5}@C$ nanocables, it is hard to understand why the $\text{MoS}_{1.5}\text{Te}_{0.5}@C$ nanotubes have an inferior performance. They both have one-dimensional and porous structure as well as similar chemical composition. There is no intrinsic difference in morphology and structure. Actually, the $\text{MoS}_{1.5}\text{Te}_{0.5}@C$ nanotubes should have advantages over nanocables in terms of volume accommodation. This manuscript needs to clarify this point. Why do we need nanocables but not nanotubes?*

Response: We appreciate the reviewer very much for his/her insightful views regarding the effects of the nanostructure. Both $\text{MoS}_{1.5}\text{Te}_{0.5}@C$ nanocables and $\text{MoS}_{1.5}\text{Te}_{0.5}@C$ nanotubes show good cycling performance at initial 100 cycles at 0.1 A g^{-1} . Nevertheless, the $\text{MoS}_{1.5}\text{Te}_{0.5}@C$ nanocables do exhibit an improved performance with good cycling stability and high rate capability. Accordingly, we supplied the associated descriptions to discuss the possible reasons for such difference between $\text{MoS}_{1.5}\text{Te}_{0.5}@C$ nanocables and $\text{MoS}_{1.5}\text{Te}_{0.5}@C$ nanotubes: 1) The nanowire inside each nanotube could dramatically increase inner solid content and provide more region for redox reaction to improve specific capacity at high rate; 2) the Na^+ ions storage behavior at high rate is mainly dominated by surface-capacitance behavior, instead of diffusion-controlled process. Both outer surface (nanotube) and inner surface (including nanowire) endow electrode materials with more surface active-sites and contact area between electrolyte/material, which can promote Na^+ adsorption and result in fast reaction kinetics; 3) The suitable nanowire-in-nanotube structure reserves sufficiently but not

excessive void between core and shells, which can effectively alleviate volume expansion and ensure favorable transport kinetics for both sodium-ion and electron during charging/discharging process. Thus, the unique properties in MoS_{1.5}Te_{0.5}@C nanocables, including abundant redox active sites, enhanced electronic conductivity and suitable buffer space are benefited for improving electrochemical performance.

Comment 2: *A large amount of S atoms are substituted by Te atoms (25%). This should have significantly effects the crystalline structure and chemical properties. How are Te atoms incorporated into the MoS₂ crystalline frameworks? More characterization should be conducted to clearly understand the doping effects.*

Response: We appreciate the reviewer very much to notice such a detail. The in-situ tellurization of MoS₂ can be implemented by the H₂Te gas (formed from the inner Te nanotube via a reaction with H₂ gas, Equation R3) with finally forming MoS_{1.5}Te_{0.5} (Equation R4) during calcining process.

We agree with reviewers' view that the introduction of Te atoms into MoS₂ lattice framework will significantly affect the crystalline structure and chemical properties of materials. The aberration-corrected high-angle annular dark-field scanning TEM image (Figure R7) of the MoS_{1.5}Te_{0.5} nanocable exhibits that there are a few brighter spots occupy some anionic sites, corresponding to heavier Te, which suggests the Te atoms have successfully been introduced in the crystalline structure of MoS₂. Besides, we can also observe that several anions that have disappeared in the selected regions (as marked by the white circles), revealing the formation of anion defect after the incorporation of Te atoms. These results are in accordance with the XRD, XPS, Raman and EPR results.

Figure R7. The HAADF-STEM images for $\text{MoS}_{1.5}\text{Te}_{0.5}@C$ nanocables

Comment 3: *In the main text, figures are not correctly cited. For example, the discussion on electrochemical performance cites Figure 1, but it should be Figure 2. “Figure 4b presents the charging-discharging curve of $\text{MoS}_{1.5}\text{Te}_{0.5}@C$ nanocables//EG SDIBs” should be “Figure 4d”. Figure 4d and Figure 5c appear before Figure 4a and Figure 5a. This is weird and un-logical. Please check.*

Response: Many thanks to the reviewer for his/her correction. We thus made a careful checking through the manuscript and have already corrected typos and errors in the revised manuscript.

Comment 4: *Figure 3b shows significant effects of electrolyte concentration on cycling performance. But no explanation on this. What’s the reason?*

Response: We appreciate the reviewer for his/her kind suggestion. In DIBs, both cations and anions as active species will participate in the energy storage. Thus, the composition parameters in electrolyte, including the solvent, the salt concentration, the type of anion, and the type of additives will influence the ions storage behavior and lead to diversity of the reversible capacity,

the coulombic efficiency, and the average discharge potential. Among these, the electrolyte concentration exerts a pivotal role in the electrode/electrolyte interfaces, which inevitably affects the performance of SDIBs.

According to the electrochemical performance of SDIBs in varied concentration electrolyte, it is proved that a high concentrated electrolyte can improve the electrochemical performance of SDIBs, which could be attributed to the stable SEI layer formation. (Please see page 14 in the revised manuscript). The compositions of SEI layer in low concentrated electrolyte (LCE) and high concentrated electrolyte (HCE) are quite different (Figure R8). In LCE, the SEI layer is mainly originated from the decomposition of solvents, which is loose and thermodynamically unstable, thus inevitably aggravating continuation of side reaction and decreasing the coulombic efficiency. However in HCE, the association between Na^+ ions and the anions can be enhanced by the high concentrated Na salts, reducing the presence of the free solvent molecules. Therefore, the SEI layer is mainly derived from by the anions of Na salts. This is thinner, more compact, and thermodynamically stable, which can effectively further suppress the decomposition of electrolyte and ensure high reversibility for ions intercalation/deintercalation, which is benefit for maintaining high coulombic efficiency and durable cycling performance of SDIBs.^[19] (Please see page 32 in the supporting information).

Figure R8. The SEI layer formation mechanism in SDIBs using (a) low concentrated electrolyte and (b) high concentrated electrolyte. (Please see the Figure S27 in page 32 from the revised supporting information)

Comment 5: *Figure 3f, the show up on the lighting diodes doesn't make sense to this work, and can be removed or moved to SI.*

Response: We appreciate the reviewer for his/her suggestion. Following the reviewer's kind suggestions, we have moved the LED photographs to the supporting information, as shown in Figure S29. (Please see page 34 in the supporting information).

Supplementary References

- (1) Jiang, C. et al. Flexible interface design for stress regulation of a silicon anode toward highly stable dual-ion batteries. *Adv. Mater.* **32**, 1908470 (2020).
- (2) a) Yang, L. et al. Tellurization velocity-dependent metallic–semiconducting–metallic phase evolution in chemical vapor deposition growth of large–area, few-layer MoTe₂. *ACS Nano* **11**, 1964 (2017); b) Zhou, L. et al. Large-area synthesis of high-quality uniform few-layer MoTe₂. *J. Am. Chem. Soc.* **137**, 11892 (2015); c) Keum, D. H. et al. Bandgap opening in few-layered monoclinic MoTe₂. *Nat. Phys.* **11**, 482 (2015); d) Qi, Y. et al. Superconductivity in Weyl semimetal candidate MoTe₂. *Nat. Commun.* **7**, 11038 (2016); Ma, N. et al. Novel 2D layered molybdenum ditelluride encapsulated in few-layer graphene as high-performance anode for lithium-ion batteries. *Small* **14**, 1703680 (2018).
- (3) Dong, S. et al. A novel coronene//Na₂Ti₃O₇ dual-ion battery. *Nano Energy* **40**, 233 (2017).
- (4) Wang, X. et al. Phosphorus-doped porous hollow carbon nanorods for high-performance sodium based dual-ion battery. *J. Mater. Chem. A* **8**, 4007 (2020).
- (5) Wang, X. et al. Regulate phosphorus configuration in high P-doped hard carbon as a super anode for sodium storage. *ACS Appl. Mater. Interfaces* **13**, 12059-12068 (2021).
- (6) Sheng, M. et al. A novel tin-graphite dual-ion battery based on sodium-ion electrolyte with high energy density. *Adv. Energy Mater.* **7**, 1601963 (2017).
- (7) Wang, X. et al. Commercial carbon molecular sieves as a Na⁺-storage anode material in dual-ion batteries. *J. Electrochem. Soc.* **164**, 3649-3656 (2017).
- (8) Dong, J. et al. Intercalation pseudocapacitance of FeVO₄·nH₂O nanowires anode for high-energy and high-power sodium-ion capacitor. *Nano Energy* **73**, 104838 (2020).
- (9) Ren, X. et al. Tailored plum pudding-like Co₂P/Sn encapsulated with carbon nanobox shell as superior anode materials for high-performance sodium-ion capacitors. *Adv. Energy Mater.* **9**, 1900091 (2019).
- (10) Zhu, Y. et al. Fast sodium storage in TiO₂@CNT@C nanorods for high-performance Na-ion capacitors. *Adv. Energy Mater.* **7**, 1701222 (2017).
- (11) Liu, B. N-Doped carbon modifying MoSSe nanosheets on hollow cubic carbon for high-

- performance anodes of sodium-based dual-ion batteries. *Adv. Funct. Mater.* 2101066 (2021).
- (12) Fan, L. et al. Soft carbon as anode for high-performance sodium-based dual ion full battery. *Adv. Energy Mater.* **7**, 1602778 (2017).
- (13) Liu, Y. et al. Layer-by-layer stacked nanohybrids of N,S-codoped carbon film modified atomic MoS₂ nanosheets for advanced sodium dual-ion batteries. *J. Mater. Chem. A* **7**, 24271 (2019).
- (14) Ma, R. et al. Offset initial sodium loss to improve coulombic efficiency and stability of sodium dual ion batteries. *ACS Appl. Mater. Interfaces* **10**, 15751 (2018).
- (15) Jiang, X. et al. A nonflammable Na⁺-based dual-carbon battery with low-cost, high voltage, and long cycle life. *Adv. Energy Mater.* **8**, 1802176 (2018)
- (16) Zhu, H. et al. Penne-like MoS₂/carbon nanocomposite as anode for sodium-ion-based dual-ion battery. *Small* **14**, 1703951 (2018).
- (17) Hu, Z. et al. All carbon dual ion batteries. *ACS Appl. Mater. Interfaces* **10**, 35978 (2018).
- (18) He, H. Et al. Anion vacancies regulating endows MoSSe with fast and stable potassium ion storage. *ACS Nano* **13**, 11843 (2019).
- (19) a) Liu, T. et al. A high concentration electrolyte enables superior cycleability and rate capability for high voltage dual graphite battery. *J. Power Sources* **437**, 226942 (2019); b) Zheng, J. et al. Research progress towards understanding the unique interfaces between concentrated electrolytes and electrodes for energy storage applications. *Adv. Sci.* **4**, 1700032 (2017); c) Heckmann, A. et al. Towards high-performance dual-graphite batteries using highly concentrated organic electrolytes. *Electrochim. Acta* **260**, 514 (2018); d) Yamada, Y. et al. Unusual stability of acetonitrile-based superconcentrated electrolytes for fast-charging lithium-ion batteries. *J. Am. Chem. Soc.* **136**, 5039 (2014).

Reviewers' comments:

Reviewer #1 (Remarks to the Author):

The authors have carried out additional experimental and theoretical work to elevate the quality of the original manuscript. However, I still have great concerns about the novelty of the work, and novelty is the top priority of assessment criteria when targeting a high-profile journal like Nat. Commun. Therefore, I recommend a rejection of this paper and encourage the authors to submit it to a more specialized journal.

Concerns about novelty: (i) Using Te nanotubes as a soft template has been a longstanding method to fabricate nanotubes, and the method has been seen in a large number of literature. To name a few, ACS Appl. Mater. Interfaces, 2018, 10, 43; J. Crys. Growth, 2009, 311, 4467; J. Alloy Compd. 2014, 617, 247. One may argue the presented work used Te as a doping source as well, but it boils down to the specific cases of materials and does not solidify novelty of the synthesis. See the example in ACS Appl. Mater. Interfaces, 2018, 10, 43, in which a similar synthetic route was used to fabricate sulfides but without Te doping (so why different in this work?). (ii) The authors also highlighted the "in-depth and systematic study" carried out in this work, but unfortunately, the contributing factors to the performance, which were concluded from the study, had nothing new from those that have been routinely seen in battery literature, such as interlayer expansion, enhanced electronic conductivity, volume buffer zone, etc. This work is a nice piece of engineering work but falls far short in terms of scientific novelty.

Comments on other issues:

- (i) The pitch on DIBs: I don't think the authors have addressed this comment. Mixing NIBs and DIBs in terms of developing high capacity anode is not appropriate. Good NIB cathodes have a significantly higher capacity than the carbons in DIBs; hence, it makes a lot more sense pairing a high-capacity anode like the one in this work with a good NIB cathode, although the voltage issue still remains.
- (ii) The advantage of Te over S: I disagree with the authors that MoTe₂ would have a higher tendency to be interlayer-expanded than MoS₂. I believe the advantage lies in the more covalent nature in transition metal-Te bonds than transition metal-S bonds, which makes the former easier to break during the conversion reaction, hence having a less overpotential.
- (iii) ICE: I don't think the authors clearly and specifically addressed this comment, by blending every possible reason together.

Reviewer #2 (Remarks to the Author):

Liu et al reported a source-template synthetic strategy for the fabrication of nanowire-in-nanotube structured MoS_{1.5}Te_{0.5}@C nanocables, which was studied as sodium ions storage materials showed potential in improving the associated electrochemical performance of sodium-based dual-ion batteries. The synthetic route is interesting and the manuscript is well-organized with sound analysis of in-situ or ex-situ characterization. It is a piece of interesting work and I would recommend its publication consideration if the authors can well address the below issues:

1. In the ex-situ XRD patterns (Figure 4C), there is an obvious "reflection splitting" phenomenon that is indicative of the GIC formation of anion intercalation. The authors are suggested to give a more detailed analysis and discussion about the working mechanism of PF₆⁻ intercalation, including, how much did the graphite layers expand? What stage GIC is formed?
2. The materials with capacitive behavior are potential candidates for high power sodium ion storage anodes. The authors should explain why the MoS_{1.5}Te_{0.5}@C nanocables show a capacitance-controlled behavior, and deliver higher capacitance-controlled contribution than the other two samples.
3. In Figure 2b, why does the CNT@MoS₂ nanotubes would deliver a higher capacity than that of the MoS_{1.5}Te_{0.5}@C nanocables in initial 100 cycles?

4. As shown in Figure 2b and 2h, both curves show the cycling stability tests for the set of anode materials. Can the former test as well reveal the long-term cycling capability as the latter test do?
5. In Figure 2f and S17, the MoS_{1.5}Te_{0.5}@C nanocables and CNT@MoS₂ nanotubes show some differences in the CV curves. The authors should give a reasonable explanation for this issue.
6. In view of rate performance, why is the performance different between MoS_{1.5}Te_{0.5}@C nanocables and MoS_{1.5}Te_{0.5}@C nanotubes so much at a large current density (5 A g⁻¹), but close to the CNT@MoS₂ nanotubes before doping, they need to explain.

Responses to reviewers' comments

Response to comments from the reviewer # 1:

General Comments. *The authors have carried out additional experimental and theoretical work to elevate the quality of the original manuscript. However, I still have great concerns about the novelty of the work, and novelty is the top priority of assessment criteria when targeting a high-profile journal like Nat. Commun. Therefore, I recommend a rejection of this paper and encourage the authors to submit it to a more specialized journal. Concerns about novelty: (i) Using Te nanotubes as a soft template has been a longstanding method to fabricate nanotubes, and the method has been seen in a large number of literatures. To name a few, ACS Appl. Mater. Interfaces, 2018, 10, 43; J. Crys. Growth, 2009, 311, 4467; J. Alloy Compd. 2014, 617, 247. One may argue the presented work used Te as a doping source as well, but it boils down to the specific cases of materials and does not solidify novelty of the synthesis. See the example in ACS Appl. Mater. Interfaces, 2018, 10, 43, in which a similar synthetic route was used to fabricate sulfides but without Te doping (so why different in this work?). (ii) The authors also highlighted the “in-depth and systematic study” carried out in this work, but unfortunately, the contributing factors to the performance, which were concluded from the study, had nothing new from those that have been routinely seen in battery literature, such as interlayer expansion, enhanced electronic conductivity, volume buffer zone, etc. This work is a nice piece of engineering work but falls far short in terms of scientific novelty.*

Response: The reviewer has done an excellent job and has offered us with valuable comments with a clear goal to make the manuscript better; we are pleased to take this opportunity to clarify the novelty and the importance of the present work.

Above all, the preparation of 1D nanotubes inevitably involve the tough etching procedures of template-removal that results in relatively high costs, tedious processes, as well as issues of environmental pollution, as summarized in Table R1. In this work, we reported a one-step strategy to fabricate wire-in-tube structured $\text{MoS}_{1.5}\text{Te}_{0.5}\text{@C}$ nanocable with Te nanowires as both template and Te sources, which allow us to implement one-stone-two-birds synthesis of various nanocables and to mitigate the above issues to a certain degree. Inspired

by the reviewer's comments, we further demonstrate the novelty of the synthetic route by conducting experiments to extend the present route to prepare a variety of metal sulfide nanocables, including wire-in-tube $WS_xTe_y@C$ and $ReS_xTe_y@C$ nanocable (Figure R1). In this context, the focus in this submission is to develop a novel and efficient synthetic route for nanocables structure with the **novelties in terms of nanostructure design and synthetic route**.

Action: We accordingly supply the associated contents to illustrate the novelties of this work: More interestingly, this method can be extended to a general approach for the preparation of the other wire-in-tube structural metal sulfide with Te-doping, including wire-in-tube $WS_xTe_y@C$ nanocable and wire-in-tube $ReS_xTe_y@C$ nanocable, which further highlight the novelty in synthetic strategy. (Please see in page 7 in the revised manuscript, and Figure S6 in the revised supporting information).

Secondly, we agree with the referee that Te nanostructures were previously reported as hard templates to prepare hollow nanostructure. Nevertheless, the present work show stark contrast to these works that only nanotube, other than nanocables, can be obtained by employing Te nanomaterials as template (Table R1). For instance, $CoTe_2$ nanotubes were synthesized by hydrothermal method with Te powder and Co-source as source (*J. Cryst. Growth*, **2009**, 311, 4467); while for the other work (*J. Alloy Compd.* **2014**, 617, 247), Te nanotubes were used as template reacting with Bi precursor to form Te/Bi_2Te_3 core/shell nanotubes. For the work (*ACS Appl. Mater. Interfaces*, **2018**, 10, 43), Te nanowires were applied as template to prepare tubular ReS_2 , additional etching step is required to remove Te template. The direct employment of Te as both templates and Te source has rarely been reported yet, especially for nanocables fabrication, which presents advantages of more convenience and effectiveness.

In addition, the contributing factors to the improved Na^+ storage performance are concluded based on the previous studies, whereas in this work, the substitutional doping of heteroatoms Te atom, and corresponding structural changes (such as expanded interlayers, anion defect, and enhanced electrical conductivity) are characterized by systematic characterization analysis (e.g. HAADF-STEM, EPR, XPS); besides, the insightful electrochemical mechanism analysis of SDIBs also are carried out by a series of systematic electrochemical tests (e.g. *ex-situ* XRD, Raman, TEM, kinetics analysis and DFT

calculations), confirming Te-doping play a vital role in improving the electrochemical performance, that's why we claim an "in-depth and systematic study" is performed to investigate the improved performance.

Last but not the least, to the best of our knowledge, this is the first report of the development of high-level Te doped MoS₂ electrode material for Na⁺-storage in SIBs and SDIBs, and we verified such Te-doping do play a key role in improving the electrochemical performance of SIBs and SDIBs: (1) The partial replacement of S by large-size Te atom in MoS₂ induce the formation of S_{1.5}Te_{0.5} interlayer ligands, resulting in the generation of abundant of defects with enhancing interlayer spacing (as evidenced by the EPR, XRD and TEM results, Figure R2), which tends to facilitate Na⁺ transportation; (2) After Te doping, the density of states around the Fermi level is strengthened, which not only improves the electric conductivity but also increases the unpaired electrons near the Fermi level, thus leading to increased reactivity and improved Na capacity (Figure R3a,c); (3) The weaker interaction of Mo-Te bonds is beneficial for the conversion reaction than that in Mo-S bands, which makes the former exhibit a higher reversibility and a lower reaction barrier (Figure R3b,d), resulting in a significantly improved electrochemical performance.

Action: Accordingly, the relative descriptions have been added and highlighted in the manuscript: As shown in Figure R3a, it should be noted that the Mo-Te bond distance in MoS_{1.5}Te_{0.5} phase (2.70 Å) is significantly longer than the Mo-S bond (2.41 Å) in MoS₂, which makes the former easier to break during the conversion reaction, hence presenting faster reaction kinetics and higher reversibility. Figure R3b performs crystal orbital Hamilton population (COHP) analyses based on the interaction of Mo-S bond in MoS₂ and MoS_{1.5}Te_{0.5}. We found the integrated COHP (ICHOP) of MoS₂ is smaller than that of MoS_{1.5}Te_{0.5} around the Fermi level, which implies the introduction of Te would be conducive to weakening the Mo-S bond of MoS_{1.5}Te_{0.5} and facilitating the bond breaking. The density of states (DOS) of MoS₂, defect-free MoS_{1.5}Te_{0.5} and defect-rich MoS_{1.5}Te_{0.5} are presented in Figure R3c. There is an obvious band gap in the bare MoS₂ model in the vicinity of the Fermi level but disappears in MoS_{1.5}Te_{0.5} systems. This result demonstrates the density of states around the Fermi level is strengthened owing to Te-doping, which not only improves the electronic conductivity, but also increases the unpaired electrons near the Fermi level, thus leading to increase the reactivity of the material, and strengthen the Na adsorption capacity. Thus, the

both ionic and electronic transport would be accelerated by Te-doping. To understand the reaction process, the Gibbs free energy evolution from the initial phase to the final phase transformation reaction is simulated (Figure R4d). Obviously, the reaction step of each electrode is downhill in free energy for transformation reaction, but it presents a more negative trend for the $\text{MoS}_{1.5}\text{Te}_{0.5}$. It means that the Na-ions storage on the $\text{MoS}_{1.5}\text{Te}_{0.5}$ electrode is more favorable during the electrochemical reaction process due to the lower free energy, thus resulting in an efficient reaction kinetics, which is in accord with the experiment results. (Please see the Figure 5 in page 30, and the page 18 in the revised manuscript).

Figure R1. The SEM images of (a,b) wire-in-tube $\text{WS}_x\text{Te}_y@\text{C}$ nanocables and (d,e) wire-in-tube $\text{ReS}_x\text{Te}_y@\text{C}$ nanocables. XRD results of (c) wire-in-tube $\text{WS}_x\text{Te}_y@\text{C}$ nanocables and (f) wire-in-tube $\text{ReS}_x\text{Te}_y@\text{C}$ nanocables. EDX spectrum of the $\text{WS}_x\text{Te}_y@\text{C}$ nanocables (g) and $\text{ReS}_x\text{Te}_y@\text{C}$ nanocables (h).

Figure R2. a) High-resolution TEM image and b) the magnified high-resolution STEM image for $\text{MoS}_{1.5}\text{Te}_{0.5}@\text{C}$ nanocables. c) XRD pattern, and d) EPR results for the $\text{MoS}_{1.5}\text{Te}_{0.5}@\text{C}$ nanocables, $\text{MoS}_{1.5}\text{Te}_{0.5}@\text{C}$ nanotubes and $\text{CNT}@\text{MoS}_2$ nanotubes.

Figure R3. a) The optimized structures of MoS_2 and $\text{MoS}_{1.5}\text{Te}_{0.5}$. b) Average pCOHP and the

corresponding integral patterns for MoS_2 and $\text{MoS}_{1.5}\text{Te}_{0.5}$. c) Total density of states (DOS) of MoS_2 , $\text{MoS}_{1.5}\text{Te}_{0.5}$, $\text{MoS}_{1.5}\text{Te}_{0.5}$ with defects. d) Free energy diagrams for the intercalation reaction and conversion reaction of MoS_2 and $\text{MoS}_{1.5}\text{Te}_{0.5}$.

Table R1. Comparison of MoS_{1.5}Te_{0.5}@C nanocables with reported Te template-based materials.

Materials	Material structure	Synthetic method	Morphology SEM image	Morphology (TEM or SEM image)	Reference
Te template-based materials					
MoS _{1.5} Te _{0.5} @C nanocables	 1) Wire-in-tube nanocable structure; 2) Few-layers (1-3 layers); 3) Expanded interlayers distance (0.75 nm) 	 1) Te nanotube as both template and Te-doping source 2) Without additional etching process 3) high-level Te-doped MoS₂ nanocables 			This work
ReS ₂ nanotubes	 1) Nanotubes; 2) Multilayered structure (>7 layers); 3) Normal interlayer spacing (0.62nm) 	 1) Te nanowires as sacrificial templates 2) Require template removal 			[1] ACS Appl. Mater. Interfaces , 2018 , 10, 43
CoTe ₂ nanotubes	Nanotube is composed of nanoparticles	 1) Te powder as Te source, CoCl₂ as Co source 2) Hydrothermal self-assembly 			[2] J. Cryst. Growth , 2009 , 311, 4467

Te/Bi ₂ Te ₃ core/shell nanotube	Nanotube	Te nanotube as hard template			[3] J. Alloy Compd. 2014, 617, 247
The electrospinning methods (Need accurate temperature control or template-removal)					
CNT/MoS ₂ hybrids	CNT-in-tube structure	 1) Electrospinning 2) CNT/PAN as hard template 3) Need remove the PAN interlayer to prepare the target architecture 			[4] Sci. Adv. 2016, 2, e1600021
Fiber-in-tube Co ₉ S ₈ -C/Co ₉ S ₈ hybrid	Wire-in-tube structure	 1) Electrospinning 2) PAN fiber as hard template 3) Need remove the PAN interlayer to prepare the target architecture 			[5] Angew. Chem. Int. Ed. 2019, 58, 6239-6243

Tube-in-tube Sn/SnO₂@Carbon	Tube-in-tube structure	 1) Electrospinning 2) Need accurate temperature control (Fast heating and quenching) 			[6] Angew. Chem. Int. Ed. 2020, 59, 2465-2472
Multi-step template methods (Need remove the secondary template)					
Tube-in-tube silicon	Tube-in-tube structure	 1) Ge nanowire as hard template 2) Need etch silicon substrates to prepare the target architecture 			[7] Angew. Chem. Int. Ed. 2009, 48, 8699-8702
Cable-Like MWNTs@SnO₂@C	Cable-like structure	 1) MWNTs as primary template, SiO₂ as secondary template. 2) Need remove the SiO₂ interlayer 			[8] Adv. Sci. 2015, 2, 1500097
Cu₂S@void@C nanotube	Tube-in-tube	 1) [Cu(Tu)₂]Cl·0.5H₂O nanowires as hard template, SiO₂ as secondary template. 3) Need etch SiO₂ interlayer 			[9] J. Mater. Chem. A, 2019, 7, 12815-12824

Comment 1. *The pitch on DIBs: I don't think the authors have addressed this comment. Mixing NIBs and DIBs in terms of developing high capacity anode is not appropriate. Good NIB cathodes have a significantly higher capacity than the carbons in DIBs; hence, it makes a lot more sense paring a high-capacity anode like the one in this work with a good NIB cathode, although the voltage issue still remains.*

Response: The reviewer raised an important issue and we appreciate him/her very much, and we are grateful to take this opportunity to clarify this point. In this work, we focused on the finding of the development of newly materials for Na⁺ storage, and DIBs were set up majorly to illustrate its (MoS_{1.5}Te_{0.5}@C nanocables) feasibility for practical application. We believed that the MoS_{1.5}Te_{0.5}@C nanocables can be well fitted for the anode of a full NIB batteries, as kindly recommended by the reviewer.

For DIBs, a desired anode of DIBs should also co-work with the cathode with matched capacity, coordinated rate capabilities, and cyclic stability between anode and cathode. Although the major issues in DIBs is the limited capacity of the cathode, a high-capacity anode is desirable to offset the potential issues of the metal plating in the practical situation for ensuring the cells' safety risks and improving lifespan. Hence, constantly exploring the high-performance negative electrode materials is also demanded.

In addition, the capacities (C) released at the cathode and the anode should be well balanced (Eq. R1), so that the cathode and the anode can be fully used. Therefore, the specific capacity based on the total mass of anode and cathode for a full cell with a fixed mass loading of electrodes can be rationally increased by **enhancing the specific capacity of anode (C_{a-s})** (Eq. R2):

$$Q_{\text{cell}} = C_{\text{c-s}} \times m_{\text{c}} = C_{\text{a-s}} \times m_{\text{a}} \quad (\text{R1})$$

$$C_{\text{T-s}} = \frac{Q_{\text{cell}}}{m_{\text{a}} + m_{\text{c}}} = \frac{C_{\text{a-s}} \times m_{\text{a}}}{m_{\text{a}} + m_{\text{c}}} = \frac{C_{\text{a-s}}}{1 + m_{\text{c}}/m_{\text{a}}} \quad (\text{R2})$$

(Where C_{a-s}, C_{c-s}, C_{T-s}, m_a, and m_c are the specific capacity (mAh g⁻¹) based on the mass (mg) of anode, cathode, and the mass of anode and cathode, respectively, Q_{cell} is the released capacity (mAh)).

In this context, the development of high-performance anode is conducive to improving the electrochemical performance of SDIBs.

Comment 2. *The advantage of Te over S: I disagree with the authors that MoTe₂ would have a higher tendency to be interlayer-expanded than MoS₂. I believe the advantage lies in the more covalent nature in transition metal-Te bonds than transition metal-S bonds, which makes the former easier to break during the conversion reaction, hence having a less overpotential.*

Response: We appreciate the reviewer very much for his/her insightful views regarding the advantage of Te over S.

Action: we supplied the associated descriptions to discuss the advantage of Te doping in the revision: (1) Tellurium element, as one member of chalcogens group, has a larger atomic radius (~1.43 Å) than sulfur (S, ~1.05 Å), which makes the valence electron of Te atom less restrained by the nuclear charge. Thus, the substitution doping of Te atoms could narrow interlayer energy band of MoS₂, which enhances the electrical conductivity of electrode; (2) An enlarged layer spacing induced by large-sized Te atom introduction could also facilitate Na⁺ migration by decreasing the Na-ion diffusion energy barrier. (3) the weak Mo-Te interaction is easier to break than that of Mo-S bands, leading to improved reversibility. Therefore, the MoS_{1.5}Te_{0.5}@C nanocable anode delivers an enhanced sodium storage property in cycling stability and rate performance in both SIBs and SDIBs. (*Please see in page 17 in the revised manuscript*)

Comment 3. *ICE: I don't think the authors clearly and specifically addressed this comment, by blending every possible reason together.*

Response: Thanks to the reviewer for his/her kind and valuable comment, and we carefully re-think this issue about ICE.

The ICE is defined as the ratio of discharge capacity to charge capacity during the initial cycle of battery. The inferior ICE depends on the irreversible capacity loss at 1st cycle, which is primarily derived from the low reversibility of the sodiation/desodiation process and the SEI layer formation. The specific surface area is one of key factors impacting ICE: (1) The large specific surface area has more oxygen-containing functional groups, which have a strong adsorption force for sodium ions, heavily hampering the migration of Na⁺ during subsequent charge process; (2) The large specific surface area tends to incur electrolyte decomposition with forming thick solid electrolyte interphase (SEI), resulting in low ICE.

The CNT@MoS₂ nanotubes display about 1.5 times higher specific surface area (36.7 m²g⁻¹) than that of MoS_{1.5}Te_{0.5}@C nanocables (21.9 m²g⁻¹), which make the former exhibit large irreversible capacity. (3) In addition, Te doping not only expand interlayer spacing but also from weak Mo-Te bonds, which is favorable for the rapid Na⁺ diffusion in bulk phase and improve reversible conversion reaction, thus enhancing the reversibility of sodiation/desodiation process. Therefore, the MoS_{1.5}Te_{0.5}@C nanocables exhibit considerably higher ICE with high reversibility for Na⁺ ions storage than that of CNT@MoS₂ nanotubes.

Response to comments from the reviewer # 2:

General Comments. *Liu et al reported a source-template synthetic strategy for the fabrication of nanowire-in-nanotube structured MoS_{1.5}Te_{0.5}@C nanocables, which was studied as sodium ions storage materials showed potential in improving the associated electrochemical performance of sodium-based dual-ion batteries. The synthetic route is interesting and the manuscript is well-organized with sound analysis of in-situ or ex-situ characterization. It is a piece of interesting work and I would recommend its publication consideration if the authors can well address the below issues:*

Response: Thank the reviewer for his/her positive and valuable comments. The reviewer brought up excellent points that are of great benefit to improve the quality of our manuscript. We have thus made substantial revision by taking into full consideration of these valuable comments. The point-by-point responses to his/her comments are presented as below.

Comment 1. *In the ex-situ XRD patterns (Figure 4C), there is an obvious "reflection splitting" phenomenon that is indicative of the GIC formation of anion intercalation. The authors are suggested to give a more detailed analysis and discussion about the working mechanism of PF6⁻ intercalation, including, how much did the graphite layers expand? What stage GIC is formed?*

Response: Thanks for the reviewer's kind suggestions.

Action: we added the associated descriptions to discuss the GIC formation of anion intercalation: According to the Bragg's Law formula, the graphite layers was expanded to 3.5 Å from the initial value (3.35 Å) when charging voltage to 4.7 V (Figure R4). The determined stages (n) in the formed PF₆⁻-GIC could be related to the periodic repeating distance (I_c), the gallery height (d_i), and the gallery expansion (Δd), according to the Eq. R3:

$$I_c = d_i + 3.35\text{Å} \times (n-1) = \Delta d + 3.35\text{Å} \times n = l \times d_{\text{obs}} \quad (\text{R3})$$

where l represents the index of (00l) oriented in the stacking direction, d_{obs} describes the observed value of the spacing between two graphitic layers, which can be calculated from diffraction angles by Bragg's Law.^[10]

Based on the calculated results (Table R2), the stage-3 GIC can be formed when the charging voltage approached to 4.7 V, implying quite a lot of PF₆⁻ anions can be intercalated in graphite space. (Please see in page 42 in the revised supporting information).

Figure R4. a) Charge-discharge curves of SDIBs marked by colorful dots in 1st cycle. b) *Ex-situ* XRD patterns of the EG cathode.

Table R2. The calculated values for the *ex-situ* X-ray diffraction measurements of charged PF₆⁻-EG electrodes of the different cell setups (the data corresponding to Figure 4c in the revised manuscript).

Cell Setup	MoS _{1.5} Te _{0.5} @C nanocables EG
d _(n+2) /d _(n+1) ratio	1.20
2θ _(n+1) (°)	25.25
2θ _(n+2) (°)	30.62
Dominant stage (n)	3
Periodic repeat distance (Ic) [Å]	14.57
PF ₆ ⁻ gallery height (d _i) [Å]	7.88
PF ₆ ⁻ gallery height (Δd) [Å]	4.53

Comment 2. *The materials with capacitive behavior are potential candidates for high power sodium ion storage anodes. The authors should explain why the MoS_{1.5}Te_{0.5}@C nanocables show a capacitance-controlled behavior, and deliver higher capacitance-controlled contribution than the other two samples.*

Response: Thanks for the reviewer's valuable comments. The surface-controlled capacitive behavior mainly occurs on the surface of electrode, which was majorly affected by electric conductivity, active sites, and specific surface area. In this work, the three samples have similar specific surface areas (Figure R5). The higher capacitive contribution for the MoS_{1.5}Te_{0.5}@C nanocables may derive from their unique nanowire-in-nanotube structure and the abundant Te doping, which can provide 1D conductive channel to assure high electric conductivity and to afford more active sites for adsorbing more Na⁺, thus the MoS_{1.5}Te_{0.5}@C nanocables electrode delivers higher capacitance-controlled contribution than the other two samples.

With increasing the scan rate, the electrochemical reactions tend to more easily occur on the surface of electrode materials in a capacitive-controlled Na-storage manner due to the requirements of fast mass transport and electron transfer rather than intercalating/de-intercalating of Na⁺ into electrode material in a diffusion-controlled Na-storage way, implying that the capacitive contribution of the MoS_{1.5}Te_{0.5}@C nanocables increases with increasing scan rate.

Figure R5. Nitrogen adsorption-desorption isothermal curves (inset: pore size distribution) for (a) MoS_{1.5}Te_{0.5}@C nanocables, (b) MoS_{1.5}Te_{0.5}@C nanotubes and (c) CNT@MoS₂ nanotubes. (d) Specific surface area and pore volume of all samples.

Comment 3. In Figure 2b, why does the CNT@MoS₂ nanotubes would deliver a higher capacity than that of the MoS_{1.5}Te_{0.5}@C nanocables in initial 100 cycles?

Response: We appreciate the reviewer for his/her insightful views on this point. It is widely reported that the MoS₂-base anode electrode delivers a higher initial capacity than the theoretical capacity of MoS₂.^[11-16] The ultra-high initial discharge capacity of CNT@MoS₂ electrode could be attributed to the following reactions: (1) in the first discharge process, the formation of SEI film will consume a large amount of Na⁺, which contributes to the specific discharge capacity. However, this capacity is irreversible; (2) the hierarchical nanostructures of materials will also affect performance, such as, enlarged interlayer spacing and defect sites both will provide more efficient storage sites for Na⁺. So, the CNT@MoS₂ nanotubes deliver a higher initial-capacity than the theoretical capacity of the MoS₂.

Comment 4. As shown in Figure 2b and 2h, both curves show the cycling stability tests for the set of anode materials. Can the former test as well reveal the long-term cycling capability

as the latter test do?

Response: Thanks for the reviewer's valuable comments. Both Figure 3c and Figure 3f show the cycling stability for anode materials. The former delivers a cycling performance at a current density of 100 mA g^{-1} while the latter at 1 A g^{-1} . At a low current density, Na^+ storage is dominated by the diffusion-controlled behavior; thus, electrode will undergo slow redox reactions for Na^+ storage. At a high rate, the electrochemical reactions tend to occur on the surface of electrode in a capacitive-controlled Na-storage manner due to the requirements of fast mass transport and electron transfer. Thus, the former cannot reveal the long-term cycling capability as the latter test.

Comment 5. *In Figure 2f and S17, the $\text{MoS}_{1.5}\text{Te}_{0.5}@C$ nanocables and $\text{CNT}@MoS_2$ nanotubes show some differences in the CV curves. The authors should give a reasonable explanation for this issue.*

Response: We would like to appreciate the reviewer very much for his/her corrections.

Action: Accordingly, the relative descriptions have been added and highlighted in the manuscript: As shown in Figure 2f and S17, one can observe that the $\text{MoS}_{1.5}\text{Te}_{0.5}@C$ nanocables show a more positive cathodic peak and a more negative anodic peak, as well as enhanced peaks intensity than that of the $\text{CNT}@MoS_2$ nanotubes electrode, implying the accelerated conversion reaction kinetics and dramatically decreased polarization of liquid-solid conversion by Te-doping.

Such significantly enhanced reaction kinetics could be ascribed to these following factors: (1) The anionic Te doping to form $\text{S}_{1.5}\text{Te}_{0.5}$ interlayer ligands narrow interlayer energy band of MoS_2 and thus achieve advanced metallic properties, improve the electrical conductivity of materials. (2) The weakened Mo-Te interaction is easier to break than that of Mo-S bands, leading to the reversible conversion reaction dynamics and reversibility. (Please see in page 20 in the revised supporting information).

Comment 6. *In view of rate performance, why is the performance different between $\text{MoS}_{1.5}\text{Te}_{0.5}@C$ nanocables and $\text{MoS}_{1.5}\text{Te}_{0.5}@C$ nanotubes so much at a large current density (5 A g^{-1}), but close to the $\text{CNT}@MoS_2$ nanotubes before doping, they need to explain.*

Response: Thank the reviewer for his/her comment. The rate capability mainly depends on the structural characteristics and electrical conductivity. The nanocable structure endows MoS_{1.5}Te_{0.5}@C nanocables with double electron transport channels and higher active sites per unit volume than the hollow nanotube structure, thus MoS_{1.5}Te_{0.5}@C nanocables deliver better rate performance than MoS_{1.5}Te_{0.5}@C nanotubes at a high rate.

It can be seen from Figure 3c, the MoS_{1.5}Te_{0.5}@C nanotubes reveals specific capacity of 411.5 mAh g⁻¹ at 0.1 A g⁻¹. Even at 5 A g⁻¹, a high specific capacity of 240.4 mAh g⁻¹ with a capacity retention rate of 57.6% can be attained. However, the CNT@MoS₂ nanotubes electrode only delivers specific capacity of 237.1 mAh g⁻¹ at 5 A g⁻¹ with an inferior capacity retention rate (45%). Obviously, the MoS_{1.5}Te_{0.5}@C nanotubes electrode exhibits better rate capacity than the CNT@MoS₂ nanotubes electrode, which could be majorly ascribed to the enhanced conductivity and enlarged interlayer spacing. It means that the enlarged interlayer distance in the MoS_{1.5}Te_{0.5}@C nanotubes, thanks Te doping, can mitigate the interlayer spacing change and provide a broad buffering-space for volume expansion to maintain structural stability during Na⁺ intercalation, which would be highly favorable for boosting the Na⁺ storage ability.

Supplementary References

- (1) Liu, S. et al. Hierarchical nanosheet-based MS₂ (M = Re, Mo, W) nanotubes prepared by templating sacrificial Te nanowires with superior lithium and sodium storage capacity. *ACS Appl. Mater. Interfaces* **10**, 37445–37452 (2018).
- (2) Li, J. et al. From Te nanotubes to CoTe₂ nanotubes: A general strategy for the formation of 1D metal telluride nanostructures. *J. Cryst. Growth*. **311**, 4467-4472 (2009).
- (3) Li, Z. et al. Rational design, high-yield synthesis, and low thermal conductivity of Te/Bi₂Te₃ core/shell heterostructure nanotube composites. *J. Alloy Compd.* **617**, 247-262 (2014).
- (4) Chen, Y. et al. Hierarchical MoS₂ tubular structures internally wired by carbon nanotubes as a highly stable anode material for lithium-ion batteries. *Sci. Adv.* **2**, e1600021, (2016)
- (5) Li, X. et al. Fiber-in-tube design of Co₉S₈-Carbon/Co₉S₈: enabling efficient sodium storage. *Angew. Chem. Int. Ed.* **58**, 6239-6243 (2019).

- (6) Gao, S. et al. A multi-wall Sn/SnO₂@carbon hollow nanofiber anode material for high-rate and long-life lithium-ion batteries. *Angew. Chem. Int. Ed.* **59**, 2465-2472 (2020)
- (7) Ishai, Mb. et al. Tube-in-tube and wire-in-tube nano building blocks: towards the realization of multifunctional nanoelectronic devices. *Angew. Chem. Int. Ed.* **48**, 8699-8702 (2009).
- (8) Zhao, Y. et al. Reserving interior void space for volume change accommodation: An example of cable-like MWNTs@SnO@C composite for superior lithium and sodium storage. *Adv. Sci.* **2**, 1500097 (2015).
- (9) Cao, G. et al. Surface chemistry of tube-in-tube nanostructured cuprous sulfide@void@carbon in catalytical polysulfide conversion. *J. Mater. Chem. A* **7**, 12815-128242 (2019).
- (10)a) Dresselhaus, M. S. et al. Intercalation compounds of graphite. *Advances in physics* **51**, 1-186 (2002); b) Rüdorff, W. et al. In situ X-ray diffraction studies of cation and anion intercalation into graphitic carbons for electrochemical energy storage applications. *Z. Anorg. Allg. Chem.* **640**, 1996–2006 (2014).
- (11) Bai, Y. et al. Neuron-inspired design of high-performance electrode materials for sodium-ion batteries. *ACS Nano*, **12**, 11503 (2018).
- (12) Pan, Q. et al. Construction of MoS₂/C hierarchical tubular heterostructures for high-performance sodium ion batteries. *ACS Nano* **12**, 12578 (2018).
- (13) Zhao, C. et al. Ultrahigh rate and long-life sodium-ion batteries enabled by engineered surface and near-surface reactions. *Adv. Mater.* 1702486 (2018).
- (14) Su, D. et al. Ultrathin MoS₂ nanosheets as anode materials for sodium-ion batteries with superior performance. *Adv. Energy Mater.* **5**, 1401205 (2015).
- (15) Choi, S. et al. 3D MoS₂-graphene microspheres consisting of multiple nanospheres with superior sodium ion storage properties. *Adv. Funct. Mater.* **25**, 1780 (2015).
- (16) Wang, G. et al. Sacrificial template synthesis of hollow C@MoS₂@PPy nanocomposites as anodes for enhanced sodium storage performance. *Nano Energy* **60**, 362 (2019)

REVIEWER COMMENTS

Reviewer #1 (Remarks to the Author):

I appreciate the authors have made extra effort to addressing my comments, on top of the additional experiments carried out in the last round of review. I agree with some parts of the response and disagree with other parts; however, I think the quality of the paper has been significantly improved compared to the initial submission and therefore, I would recommend a minor revision to address a few more comments.

1. I think the authors need to fully justify the novelty of the synthesis in the paper. Particularly when comparing with previous work using Te nanotubes as a soft template, the appraisal of this work needs to be more balanced. For instance, in Table R1. The ReS₂ nanotube work needed a step of template removal because the aimed structure wasn't tube-in-tube, which doesn't mean the synthesis was tedious. The synthesis in this work would have needed a second step if Te needed to be removed.
2. Again, I don't agree with the authors about the analysis of ICE, which is overly simplified and not much different from the response from the first revision. I strongly suggest the authors carrying out extra experiment to address this issue. The surface area difference is not significantly at all. Both hierarchical nanostructures have defect sites (re: response to comment 3 made by reviewer 2). One approach to find it out is to get rid of carbon and see how ICE looks like between the two materials. Can the authors do it? I also wonder if there is a reduced polysulfide dissolution in doped material, which might explain the continuous capacity decay of MoS₂/CNT (re: response to comment 3 made by reviewer 2). Can the authors explore it?
3. The authors should carefully appraise the performance in the paper and avoid any possible over-rating. Overrated performance has become an issue nowadays and I believe a well balanced comparison will help to put the work in context with previous work.
4. "A high-capacity anode is desirable to offset the potential issues of the metal plating in the practical situation". I don't think this is correct. It is the high voltage of an anode that offsets the issue of metal plating. Obviously, the authors need to justify in the introduction why a dual-ion battery was chosen as a demonstrator.

Reviewer #2 (Remarks to the Author):

In this revision, the authors did a good job, by conducting additional experiments and making substantial revision, to address the comments raised by the reviewers, and my concerns was well addressed. I also take time to study the reviewer1#'s comments and the associated response and the corresponding revision: 1) the novelty regarding the synthesis is improved by extending it to a general route to a set of nanocables, which may be conducive to find more functional nanocables; 2) the authors perform additional DFT and characterization to expound the improved performance in MoS_{1.5}Te_{0.5}@C Nanocables, which provide more evidences to verify the unique properties of nanocables; 3) although there remains debate on the calculation of the energy density, the authors have supplied the energy density based on mass of cathode, anode, and both. Therefore, I would be pleased to recommend its acceptance in Nat Commun after two rounds of review and revisions.

Responses to reviewers' comments

Response to comments from the reviewer # 1:

Comment 1. *I think the authors need to fully justify the novelty of the synthesis in the paper. Particularly when comparing with previous work using Te nanotubes as a soft template, the appraisal of this work needs to be more balanced. For instance, in Table R1. The ReS₂ nanotube work needed a step of template removal because the aimed structure wasn't tube-in-tube, which doesn't mean the synthesis was tedious. The synthesis in this work would have needed a second step if Te needed to be removed.*

Response: We are grateful to the referee for his/her kind and valuable suggestion. Accordingly, the novelty of the synthesis was further discussed in the revised manuscript: The Te nanomaterials (e.g. nanorod and nanowire) have been widely used as templates for fabricating a variety of tubular nanomaterials, yet there is little report for nanocables synthesis. In the previous works, the targeted materials tend to grow on the outside surface of the Te nanowire/nanotube, likely due to the diffusion limit of ions under the conventional hydrothermal process. The stirring assistance during hydrothermal reaction can facilitate the mass transportation and thus increase reaction rate as well as the flowability of solution. Therefore, it is possible for the active ions to permeate into the inner space of Te nanotube using the stirring-hydrothermal method, which ensures the formation of wire-in-tube MoS_{1.5}Te_{0.5}@C nanocables (please see page 5 in the revised manuscript).

The morphology and structure of the as-prepared Te@C/MoS₂ sample using different hydrothermal method were shown in the Figure R1, respectively. By using the conventional hydrothermal method, MoS₂ nanosheets can only grow on the outside surface of Te nanotube, as evidenced by the hollow inner part in the axial direction (Figure R1a, b). However, the nanosheets can grow on both the outer surface and the inner cavity of the nanotube in stirring-hydrothermal condition (Figure R1c, d), verifying the role of stirring in the formation of nanocables structure.

The formation mechanism of nanowire-in-nanotube MoS_{1.5}Te_{0.5}@C nanocables and the associated discussion has been supplied in the revised manuscript: The formation

mechanism of nanowire-in-nanotube $\text{MoS}_{1.5}\text{Te}_{0.5}@\text{C}$ nanocables is schematically illustrated in Figure 1a and Figure R2 or Supplementary Figure 1. Te nanotubes (Supplementary Figure 2) as the template were evenly dispersed in the mixed solution containing sodium molybdate, thiourea and glucose, the subsequent stirring-hydrothermal process induce the in-situ growth of carbonaceous film modified MoS_2 nanosheets on both inner and outer surface of Te nanotubes ($\text{Te}@\text{C}/\text{MoS}_2$, Supplementary Figure 3). The nanowire-in-nanotube structured $\text{MoS}_{1.5}\text{Te}_{0.5}@\text{C}$ nanocables can be finally evolved by annealing $\text{Te}@\text{C}/\text{MoS}_2$ in H_2/Ar atmosphere, during which Te nanotube templates are vaporized as Te source (H_2Te) for partially substitution of S in MoS_2 to form $\text{MoS}_{1.5}\text{Te}_{0.5}$ with carbon film decoration (Please see between page 5-6 in the revised manuscript). Besides, additional experiments were conducted to demonstrate that the present synthetic method can be extended to be a universal route for the preparation of various wire-in-tube structural metal sulfide composites.

Figure R1. The SEM images of $\text{Te}@\text{C}/\text{MoS}_2$ sample under a conventional hydrothermal method (a,b) and a stirring-hydrothermal method (c,d).

Figure R2. Schematic illustration of the formation process of nanotube and nanocable structures under normal and stirring hydrothermal processes (Step 1) at 180 ° C for 10 h respectively.

Comment 2. *Again, I don't agree with the authors about the analysis of ICE, which is overly simplified and not much different from the response from the first revision. I strongly suggest the authors carrying out extra experiment to address this issue. The surface area difference is not significantly at all. Both hierarchical nanostructures have defect sites (re: response to comment 3 made by reviewer 2). One approach to find it out is to get rid of carbon and see how ICE looks like between the two materials. Can the authors do it? I also wonder if there is a reduced polysulfide dissolution in doped material, which might explain the continuous capacity decay of MoS₂/CNT (re: response to comment 3 made by reviewer 2). Can the authors explore it?*

Response: We appreciate the reviewer for his/her insightful comments. Following these comments and kind suggestions, we conducted additional experiments to study if carbon film affect the role of Te-doping in influencing the initial coulombic efficiency (ICE) by preparing two samples of MoS_{1.5}Te_{0.5} and MoS₂ samples (without carbon). As

expected, $\text{MoS}_{1.5}\text{Te}_{0.5}$ shows a higher ICE than MoS_2 . Therefore, high level Te-doping in MoS_2 can indeed enhance the ICE. The associated discussion and descriptions have been supplied in the revised manuscript: In order to exclude the influence of carbon films, we carried out experiments to prepare the pure $\text{MoS}_{1.5}\text{Te}_{0.5}$ and MoS_2 samples without carbon coating and evaluate their ICE for Na^+ storage. The $\text{MoS}_{1.5}\text{Te}_{0.5}$ electrodes show an average ICE of $\sim 88.4\%$, this value is higher than that of MoS_2 sample ($\sim 74.8\%$, Figure R3a, b). Thus, the enhanced ICE of $\text{MoS}_{1.5}\text{Te}_{0.5}@C$ nanocables can be rationally attributed to improved reversibility of the sodiation/desodiation process, since the weaker interaction of Mo-Te bonds was beneficial for the conversion reaction than that of Mo-S bands, which favor for lowering reaction barrier with improved reversibility. (please see page 11 in the revised manuscripts). Besides, according to the crystal orbital Hamilton population (COHP) analyses results (Figure R3c), we found the integrated COHP (ICHOP) of MoS_2 is smaller than that of $\text{MoS}_{1.5}\text{Te}_{0.5}$ around the Fermi level, which implies the introduction of Te would be conducive to weakening the Mo-S bond of $\text{MoS}_{1.5}\text{Te}_{0.5}$ and facilitating the bond breaking. As shown in the density of states curves (Figure R3d), the $\text{MoS}_{1.5}\text{Te}_{0.5}$ displays the higher density of states (DOS) than that of MoS_2 around the Fermi level, exhibiting enhanced electron mobility during sodiation process, and improving the reversibility of intercalation and conversion reaction. Above analysis presents the improved reversibility of $\text{MoS}_{1.5}\text{Te}_{0.5}$ electrode, supporting the high ICE result.

For the second point, we also conducted experiments to examine if there is a reduced polysulfide dissolution in doped material. To this end, the glass fiber separator after 200 cycling tests at 0.1 A g^{-1} was analyzed (Figure R4a). One can observe that the color of the separator from $\text{MoS}_{1.5}\text{Te}_{0.5}@C$ nanocables is light yellow, while the $\text{CNT}@MoS_2$ nanotube shows dark yellow, implying a reduced polysulfide dissolution in Te-doped materials. In addition, the separators were further characterized by the UV-vis absorption measurement (Figure R4b), the separator of doped material exhibits much weaker polysulfide absorption peaks compared to the free-doped one. These results have evidenced that Te-atom doping could significantly enhance the reversibility of the conversion reaction and suppress the polysulfide dissolution in sodiation/desodiation

process, which contributes to the improvement in coulombic efficiency and long-lasting cycle stability.

Figure R3. a) Charge-discharge curves at 1st cycle and b) ICE of MoS_{1.5}Te_{0.5} and MoS₂ from five cells. c) Average pCOHP and the corresponding integral patterns for MoS₂ and MoS_{1.5}Te_{0.5}. d) Total density of states (DOS) of MoS₂, MoS_{1.5}Te_{0.5}, MoS_{1.5}Te_{0.5} with defects.

Figure R4. a) The digital photo of the glass fiber separator and b) the UV-vis spectra

of the electrolyte solution after adding glass fiber separator.

The glass fiber separator was gained from the cell after the cycle performance test at 0.1 A g^{-1} , which was carefully dis-assembled inside an argon-filled glove box. For the UV-vis absorption test, the obtained glass fiber separator was immersed in the electrolyte solution for 4h, and the solution was subsequently used for testing.

Comment 3. *The authors should carefully appraise the performance in the paper and avoid any possible over-rating. Overrated performance has become an issue nowadays and I believe a well-balanced comparison will help to put the work in context with previous work.*

Response: Thanks for the reviewer's kind suggestions. We totally agree that we should not over-rate the performance, especially by comparing with the previous works. Accordingly, we already made a corresponding revision in the revised manuscript by removing some unsuitable words like "superior, outstanding and novel" etc.

Comment 4. *"A high-capacity anode is desirable to offset the potential issues of the metal plating in the practical situation". I don't think this is correct. It is the high voltage of an anode that offsets the issue of metal plating. Obviously, the authors need to justify in the introduction why a dual-ion battery was chosen as a demonstrator.*

Response: We agree the reviewer's opinion. Accordingly, we modified the corresponding descriptions about dual-ion battery in the introduction: Lithium-ion batteries (LIBs) play important roles in modern society yet face challenges of the growing cost of lithium-based resources and uneven geographical distribution.¹⁻² Thus, it is widely accepted that sodium-ion batteries (SIBs) are promising alternatives to LIBs, especially in the grid-scale energy storage due to the abundant source and low cost of sodium.³ Among the emerging sodium-ion energy storage devices, sodium-based dual-ion batteries (SDIBs) with graphite as both anode and cathode have attracted increasing research interests thanks to its high operating voltage.⁴ However, the current SDIBs are still unsatisfactory to meet the requirement of high energy density, because graphite has a relatively low capacity of around 30 mAh g^{-1} for NaC_{64} as anode and about 112 mAh g^{-1} for C_{20}PF_6 as cathode, respectively.⁵ Especially for anode, soft carbon and hard

carbon have been widely studied as anode of SDIBs while show a relatively low capacity so far.⁶⁻⁷ Alloys and metal oxides with enhanced capacity have thus been investigated but tend to undergo extra volume expansion during sodiation/de-sodiation with unsatisfactory cycling stability.⁸⁻⁹ In this context, we still need to strive to explore appropriate anode materials with high capacity and stable structures, which is of great importance to push forward the prosperity of SDIBs. (Please see page 3 in the revised manuscript).

Response to comments from the reviewer # 2:

General Comments. *In this revision, the authors did a good job, by conducting additional experiments and making substantial revision, to address the comments raised by the reviewers, and my concerns was well addressed. I also take time to study the reviewer I#'s comments and the associated response and the corresponding revision: 1) the novelty regarding the synthesis is improved by extending it to a general route to a set of nanocables, which may be conducive to find more functional nanocables; 2) the authors perform additional DFT and characterization to expound the improved performance in $\text{MoS}_{1.5}\text{Te}_{0.5}@C$ Nanocables, which provide more evidences to verify the unique properties of nanocables; 3) although there remains debate on the calculation of the energy density, the authors have supplied the energy density based on mass of cathode, anode, and both.*

Therefore, I would be pleased to recommend its acceptance in Nat Commun after two rounds of review and revisions.

Response: We are grateful to the reviewer for taking time to review the revision with acceptance recommendation.

REVIEWERS' COMMENTS

Reviewer #1 (Remarks to the Author):

The authors have address the questions I raised in the last revision and therefore, I think the manuscript can be accepted in the current form.